# Expanding the active charge carriers of polymer electrolytes in lithium-based batteries using an anion-hosting cathode

Zongjie Sun[1,6], Kai Xi [1,6], Jing Chen[1], Amor Abdelkader [2], Meng-Yang Li[1], Yuanyuan Qin[1], Yue Lin[3], Qiu Jiang[4], Ya-Qiong Su[1], R. Vasant Kumar [5] & Shujiang Ding [1✉]

Ionic-conductive polymers are appealing electrolyte materials for solid-state lithium-based batteries. However, these polymers are detrimentally affected by the electrochemically-inactive anion migration that limits the ionic conductivity and accelerates cell failure. To circumvent this issue, we propose the use of polyvinyl ferrocene (PVF) as positive electrode active material. The PVF acts as an anion-acceptor during redox processes, thus simultaneously setting anions and lithium ions as effective charge carriers. We report the testing of various Li||PVF lab-scale cells using polyethylene oxide (PEO) matrix and Li-containing salts with different anions. Interestingly, the cells using the PEO-lithium bis(trifluoromethanesulfonyl)imide (LiTFSI) solid electrolyte deliver an initial capacity of 108 mAh g$^{-1}$ at 100 µA cm$^{-2}$ and 60 °C, and a discharge capacity retention of 70% (i.e., 70 mAh g$^{-1}$) after 2800 cycles at 300 µA cm$^{-2}$ and 60 °C. The Li|PEO-LiTFSI|PVF cells tested at 50 µA cm$^{-2}$ and 30 °C can also deliver an initial discharge capacity of around 98 mAh g$^{-1}$ with an electrolyte ionic conductivity in the order of 10$^{-5}$ S cm$^{-1}$.

[1] School of Chemistry, Engineering Research Center of Energy Storage Materials and Chemistry for Universities of Shaanxi Province, State Key Laboratory for Mechanical Behavior of Materials, Xi'an Jiaotong University, 710049 Xi'an, China. [2] Faculty of Science and Technology, Bournemouth University, Talbot Campus, Fern Barrow, Poole BH12 5BB, UK. [3] Cavendish Laboratory, University of Cambridge, Cambridge CB3 0HE, UK. [4] School of Materials and Energy, University of Electronic Science and Technology of China, 611731 Chengdu, China. [5] Department of Materials Science and Metallurgy, University of Cambridge, Cambridge CB3 0FS, UK. [6] These authors contributed equally: Zongjie Sun, Kai Xi. ✉email: dingsj@mail.xjtu.edu.cn

The safety requirement for lithium-based batteries and the demand for lithium metal anode have prompted researchers to look for solid-state alternatives to organic solvent-based non-aqueous liquid electrolytes[1–4]. Since Wright and coworkers reported the ionic conductivity of the mixture of polyethylene oxide (PEO) and alkali metal salts[5,6], solid polymer electrolytes (SPEs) are attractive for constructing all-solid-state lithium secondary batteries due to the lack of leakage risk[7]. Recently works have addressed the flammability problem of such SPEs using different strategies[8,9]. Yet, the weak polymer chain motion leads to insufficient ion transport for SPEs[3,10]. Therefore, many investigators have committed to developing high ion conductive SPEs. Typical strategies include designing the polymer segment structure[11–13] and combining polymer matrix with ceramic[14,15] or inorganic solid electrolytes[16,17].

One critical problem that has to be solved for constructing advanced SPEs is the movement of anions. In PEO, the repeat unit (-$CH_2$-$CH_2$-$O$-) shows chelating complexation with $Li^+$[18]. Specifically, the PEO chains can adopt a helical conformation that presents the optimum distances for O-Li interactions similar to the crown structures[19]. The large solvation structure significantly slows the migration of $Li^+$[20,21]. In contrast, the non-coordinating anions contribute most of the ionic conductance in SPEs, meaning that the majority of ion migration is irrelevant to energy generation. The facile movement of the anions and the scarce supply of cations facilitate the uneven deposition of Li onto surface protrusions, leading to the self-amplification process of dendritic growth[22–24]. Another issue caused by anion migration is concentration polarization on the electrode surface during cycles[25,26]. The polarization results in critical performance degradation, such as increasing internal resistance and decreasing operating voltage of the cell[27]. Coupling with the low $Li^+$ migration increases the electrode/solid electrolyte interface resistances significantly to accelerate the deterioration of anodes[28–30]. The short-circuit in polymer-based solid-state batteries generally predates the combination of liquid components and commercial separators, which runs counter to the original intention of solid-state electrolytes. Single-ion conducting solid polymer electrolytes (SISPEs) was suggested to alleviate the problems caused by anion migration[31,32]. By constructing polymer segments with weak interaction with $Li^+$[33] or grafting anions to the polymer backbone[34], SISPEs can achieve a high lithium-ion transference number (>0.9). Though some studies have proved that SISPEs have less strict requirements for ion conductivity[32,35], there are still concerns about the number of effective carriers due

to strong electrostatic interaction between $Li^+$ and the negative charges.

The participation of anions in the storage process has promoted the development of dual-ion batteries in the liquid or extended gel phase of the electrolyte[36–38]. These systems were intended as low-cost alternatives to lithium-ion batteries, focusing on critical parameters such as sustainability and material availability[39]. Apart from the electrochemical stability requirement, the electrolyte of dual-ion batteries is similar to that of lithium-ion batteries. The benefit of the effective carrier expansion is hidden by the abundant ionic conductivity of the liquid/gel electrolyte. However, for solvent-free SPEs with limited ion movement, the strategy of enhancing the correlation between ion migration and electrode reaction is expected to play a more crucial role. In this work, to circumvent the SPEs issue of non-reactive anion migration, we adopted an anion acceptor cathode, polyvinyl ferrocene (PVF), previously reported in liquid energy storage system[40] and matched it with PEO-based solid polymer electrolytes, and realized dual-ion solid-state batteries with long-cycle stability and high current-carrying capacity. The ferrocene units, anchored to the long-chain polymer, encourage anions as the effective charge carrier, together with $Li^+$. The expansion of carriers significantly improves the current-carrying capacity of unmodified SPEs (Fig. 1) and avoids the short-circuit failure led by concentration polarization. The Li|SPE|PVF cell maintained more 4000 cycles at 60 °C, 300 μA cm$^{-2}$. Besides, the impact of anion species on ion mobility and interaction with the cathode is also investigated in-depth, and the results proves that the electrode potential was closed related with anion species and ion aggregation state.

**Materials design and characterization.** Sustainability, high capacity, and adjustable redox properties give organic electrodes great potential[41,42]. To avoid the diffusion of active units in a non-flow system, we anchored the anion-hosting unit to the polymer chain by free-radical polymerization of vinyl ferrocene with 2,2'-azoisobutyronitrile (AIBN) as the initiator in toluene, as illustrated in Fig. 2a. The electronic structure of the cyclopentadiene and iron atom hybrid orbital provides ferrocene with stable and reversible redox properties (Fig. 2b)[43]. The energy dispersive X-Ray (EDX) elemental mapping (Fig. 2a) proves the homogeneous distribution of Fe across the PVF. The polymerization process was confirmed by the Fourier transform infrared (FT-IR) spectroscopy in Fig. 2c, which shows a weakening of double bond vibration peaks at 1625 cm$^{-1}$ after polymerization. The gel permeation chromatography (GPC) results

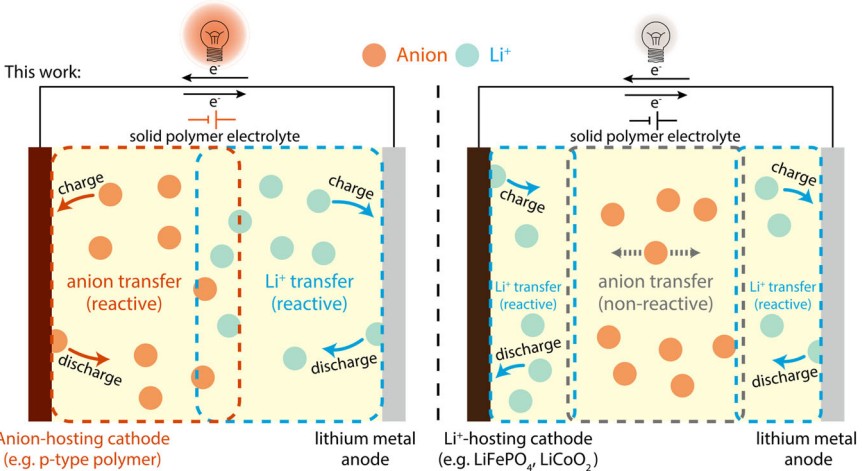

**Fig. 1 Anion and cation migration and electrode reaction in two different battery systems.** The ions transfer in SPEs of all-solid-state batteries using different ion acceptors as the cathode (left: anion-hosting cathode (this work), e.g., p-type polymer, right: $Li^+$-hosting cathode).

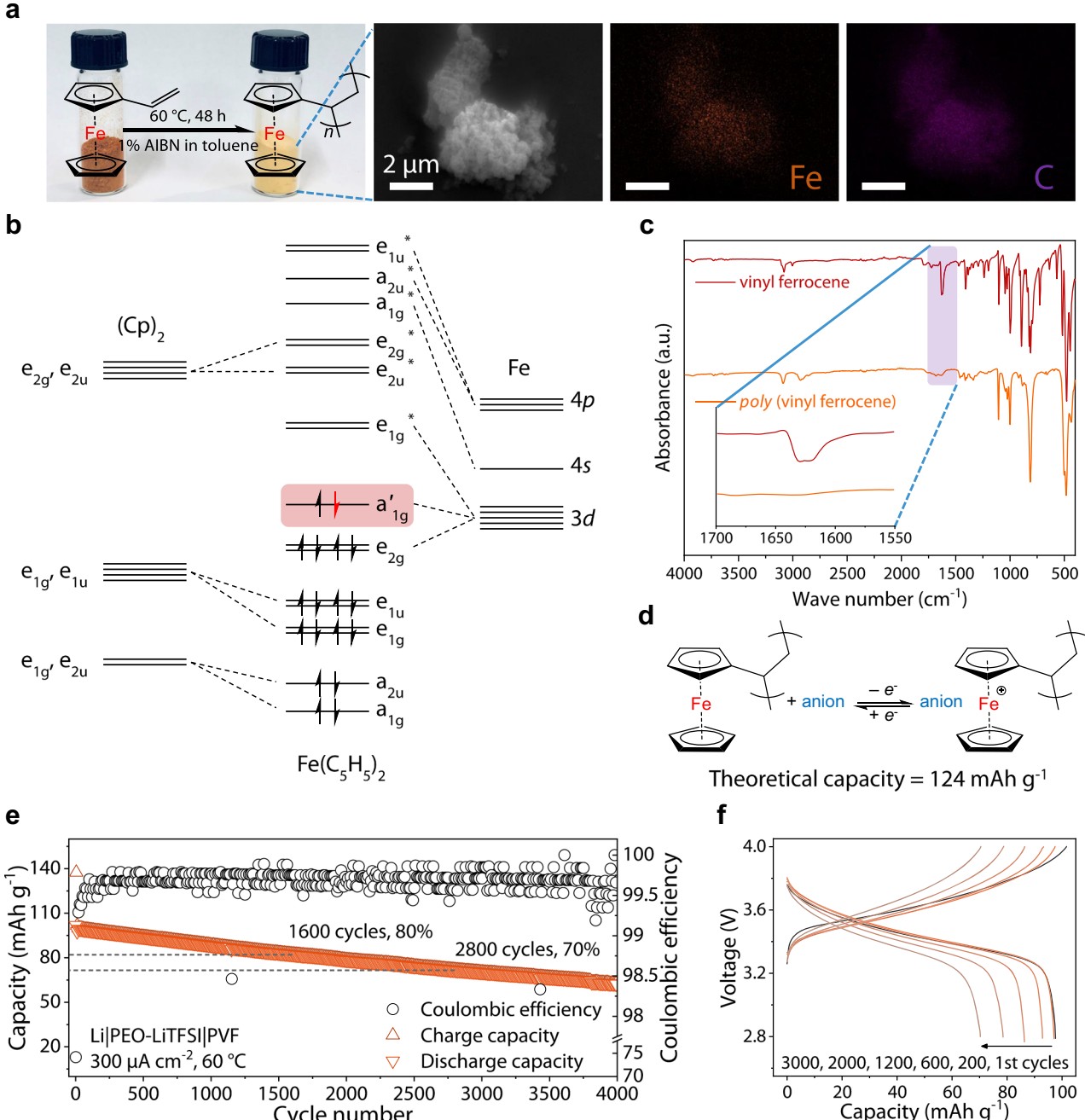

**Fig. 2 Synthesis, physicochemical and electrochemical characterization of the polyvinyl ferrocene. a** The optical image, SEM image and EDX elemental mapping results of the polyvinyl ferrocene. **b** The frontier orbital electronic structure of ferrocene/ferrocenium, while the electron transfer in the redox reaction is marked in red. **c** FT-IR spectroscopy of vinyl ferrocene and polyvinyl ferrocene, the inset shows the weakening of the peak intensity at 1625 cm$^{-1}$. **d** The redox reaction of polyvinyl ferrocene with anions in electrolyte. The theoretical capacity is based on the molecular weight of the active unit and the charge transfers numbers. **e, f** show the capacity retention and charge-discharge curves of Li||PVF cell at 60 °C, 300 μA cm$^{-2}$, respectively. The cell was cycled at 50 μA cm$^{-2}$ in the first 5 cycles.

(Supplementary Table 1) prove that PVF has a high molecular weight (~4800 g mol$^{-1}$) with wide distribution (weight-average molecular weight/number-average molecular weight, $M_w/M_n$ = 1.71). The redox activity of PVF can provide a theoretical capacity of 124 mAh g$^{-1}$ (Fig. 2d), located at the top level of the p-type organic cathode[41]. Moreover, the theoretical redox potential of ~3.45 V vs. Li$^+$/Li can be tolerated by most SPEs. The thermogravimetric analysis (TGA) in Supplementary Fig. 1 shows that PVF did not undergo significant thermal weight loss or phase change below 300 °C, ensuring the electrode stability in the case of high

temperature operation. The scanning electron microscope (SEM) and EDX elemental mapping results of the electrode show the evenly mixed PVF and conductive carbon without agglomeration (Supplementary Fig. 2).

Standard SPEs are composed of lithium salt and polymer with solvation ability. Benefiting from the high dielectric constant and chain flexibility, PEO is one of the most widely studied polymers matrix[6,7]. Based on the well-designed anion hosting positive material, the Li||PVF cell matched with the PEO matrix and lithium bis(trifluoromethanesulfonyl)imide (LiTFSI) exhibits good cycle

stability. It maintained 70% capacity retention after 2800 cycles and delivering around 65 mAh g$^{-1}$ after 4000 cycles at 300 μA cm$^{-2}$ and 60 °C (Fig. 2e). Figure 2f shows the potential profile of the cell during cycling, with a gradual decrease in the discharge potential accompanied by a rise in polarization. This can be attributed to the deterioration of ion transport caused by the diffusion of active material into the electrolyte during long-term cycling. Since the migration of anions and cations in the present work is related to the electrodes' reaction[44,45], we set lithium perchlorate (LiClO$_4$), lithium bis(oxalato)borate (LiBOB) and lithium bis(fluorosulfonyl)imide (LiFSI) in SPEs to obtain a deep insight into the anion electrode reaction. The ionic conductivity of the SPEs was evaluated through alternate current (AC) impedance (seen in Supplementary Fig. 3), measured by sandwiching the SPEs between two stainless steel electrodes. Supplementary Fig. 4 shows the ionic conductivity as a function of temperature. Owing to the delocalized negative charge of anions, PEO-LiTFSI displays the highest ionic conductivity (3.53 × 10$^{-4}$ S cm$^{-1}$ at 333 K). Except for LiClO$_4$, other electrolytes have comparable conductivity at high temperature. The differential scanning calorimetry (DSC) results in Supplementary Fig. 5 show that, compared to pure PEO, each SPEs exhibit decreased melting point and produce glass transition processes, implying enhanced segmental motion. PEO-LiTFSI and PEO-LiBOB, which exhibit high ionic conductivity, have a lower degree of crystallinity (determined by the melting peak intensity). The melting point and glass transition temperature changes with anions species are not directly related to the ionic conductivity. Different anions impact the polymer-salt composite structure, resulting in various degrees of segment motion. Meanwhile, salt dissociation affects the conductivity. The negative charge delocalization of anion helps release more free ions.

Inefficient utilization of ions in SPEs severely restrict the batteries' performance. To explore the ion mobility influence in the designed systems, we measured the lithium-ion transference number ($t_{Li+}$) of different SPEs through the steady-state current method[46,47] (results are shown in Supplementary Fig. 6, Supplementary Table 2). As seen in Fig. 3a, the electrolytes with LiTFSI, LiFSI and LiClO$_4$ as salts pose low $t_{Li+}$, which are all-around 0.1, proving that the anions contribute the most ion movement in these SPEs. Note that the calculated $t_{Li+}$ of PEO-LiBOB is −0.38, which can be attributed to the ionic aggregates[48,49]. As shown in Fig. 3b, while the negatively charged ion clusters dominate the charge transfer in the electrolyte, the $t_{Li+}$ of SPEs can be low to negative due to the short-range interactions[50,51]. The Nyquist plots of Li||PVF cells shown in Fig. 3c proves that negatively charged clusters significantly increase the cells' charge transfer resistance. As shown by the fitting results of electrochemical impedance spectroscopy (EIS) in Supplementary Table 3, the main charge transfer impedance in Li|PEO-LiBOB|PVF (971.1 Ω) is significantly higher than that in cells with TFSI$^-$ (99.32 Ω), FSI$^-$ (103.9 Ω), and ClO$_4^-$ (195.2 Ω) as anions in electrolyte. Since negative $t_{Li+}$ values often occur in high salt concentration measurements systems[49,50], the low salt concentration was conducted to prove this hypothesis. As shown in Supplementary Fig. 7, due to the positive $t_{Li+}$, PEO-LiBOB (O:Li = 30:1, 40:1) with low ionic conductivity show lower resistance. Generally, for a typical Li$^+$-hosting cathode, SPE with low $t_{Li+}$ has strict requirements on ion conductivity, especially under high current density, the scarcity of cations on the electrode surface accelerates the growth of dendrites[22]. The Li||PVF batteries assembled with SPEs all exhibit reversible charge-discharge process as shown in Fig. 3d, and the anion species lead a difference on the operating voltage of batteries. This finding prompted us to further study of anion impact.

**Anion impact on electrode reaction**. The electrochemical stability of the SPEs was evaluated using Li|SPE|stainless steel cells.

The linear scanning voltammetry (LSV) curves in Supplementary Fig. 8 show a low redox current within a voltage window up to 4.0 V vs. Li$^+$/Li, confirming that electrolyte oxidation does not disturb the electrode reaction. The electrochemical behavior of PVF with anions was evaluated through cyclic voltammetry (CV) tests. As shown in Fig. 4a, the PVF cathode shows good redox reversibility provided by the active unit. Notably, the unit's stabilization into polymer plays a crucial role in electrode reaction reversibility. The solid-state cell with the ferrocene cathode (Supplementary Fig. 9) exhibits an oxidation peak only in the first CV cycle. This can be attributed to the ion pairs (ferrocenium/ anions) diffusion, owing to SPEs' segment motion at high temperature. In addition, the poor electron/ion transfer interface between the ferrocene active material and the electrolyte further hinders the redox process. In contrast, PVF with anchored active units can maintain well reversible redox. The peak potential separation (ΔE) is informative of the electrochemical reaction kinetics. Judging from CV results, PEO-LiTFSI, has the lowest potential gap (ΔE = 0.103 V), while PEO-LiClO$_4$ has the highest, corresponding to the ionic conductance trend (Supplementary Fig. 10, Supplementary Table 4). The shape difference on the CV can be attributed to active unit (ferrocene) packing and intermolecular interactions (see Supplementary Note 1 on non-ideal behavior).

The influence of anion species on the electrode reaction in liquid electrolytes, related to anion couples, was previously investigated by Redepenning et al[52]. The results show that the ion pairs' formation could negative shift the electrode potential (E$_0$) from theoretical, affected by binding capability[53,54]. We calculate the binding energy (BE) of ion pairs by density functional theory (DFT) simulations. The cathode was simplified by substituting ethyl ferrocenium for the PVF (Supplementary Table 5). As Fig. 4b shows, the ethyl ferrocenium has the highest BE to ClO$_4^-$ and decrease with the order FSI$^-$, BOB$^-$, and TFSI$^-$. However, in the CV results of PVF, the E$_0$ with TFSI$^-$, BOB$^-$, and FSI$^-$ as anions have no significantly differences (3.463, 3.470, and 3.476 V, respectively). The discrepancy between the experimental and computational results can be explained by the steric hindrance of active units (ferrocene) and ions/ion clusters: (1) The folded long chain in PVF inhibits the binding of large anion to ferrocene, thus, reducing the BE effect on E$_0$. When using low steric hindrance for ferrocene as the cathode (Supplementary Fig. 11), the E$_0$ of TFSI$^-$, FSI$^-$ decreased significantly (Supplementary Fig. 12a), matching the trend of the calculated results; (2) The anion-dominated ion clusters exhibit a larger steric structure, which enhances the hindrance effect. In PEO-LiBOB, the change in salt concentration resulted in different aggregation morphologies of ions, as previously described (Fig. 3b, Supplementary Fig. 7). Compared with O:Li at 20:1, the E$_0$ decrease of low ferrocene steric hindrance is more evident in low salt concentration (30:1, 40:1), shown in Supplementary Fig. 12b. The results demonstrate that, apart from the cathode, the steric hindrance on the anion side also weakens the E$_0$ drop caused by the binding process. Therefore, avoiding ion clusters, ClO$_4^-$, with the smallest size (Supplementary Fig. 13) and the strongest BE (Fig. 4c), shifts E$_0$ to more negative values (3.381 V vs. Li$^+$/Li), while the other anions were not significantly affected by the binding effect (Fig. 4d).

Before further examining the all-solid-state batteries, we evaluated the capacity and stability of the PVF electrode with liquid electrolyte. As shown in Supplementary Fig. 14a, b, the capacity increases in the first few cycles (1.0 M LiTFSI in tetraglyme), which can be attributed to the cathode electrochemical activation, previously observed in several organic electrodes[55]. The electrode exhibits high initial capacity (107 mAh g$^{-1}$) and retention at 100 mA g$^{-1}$, 30 °C. Also, when lithium

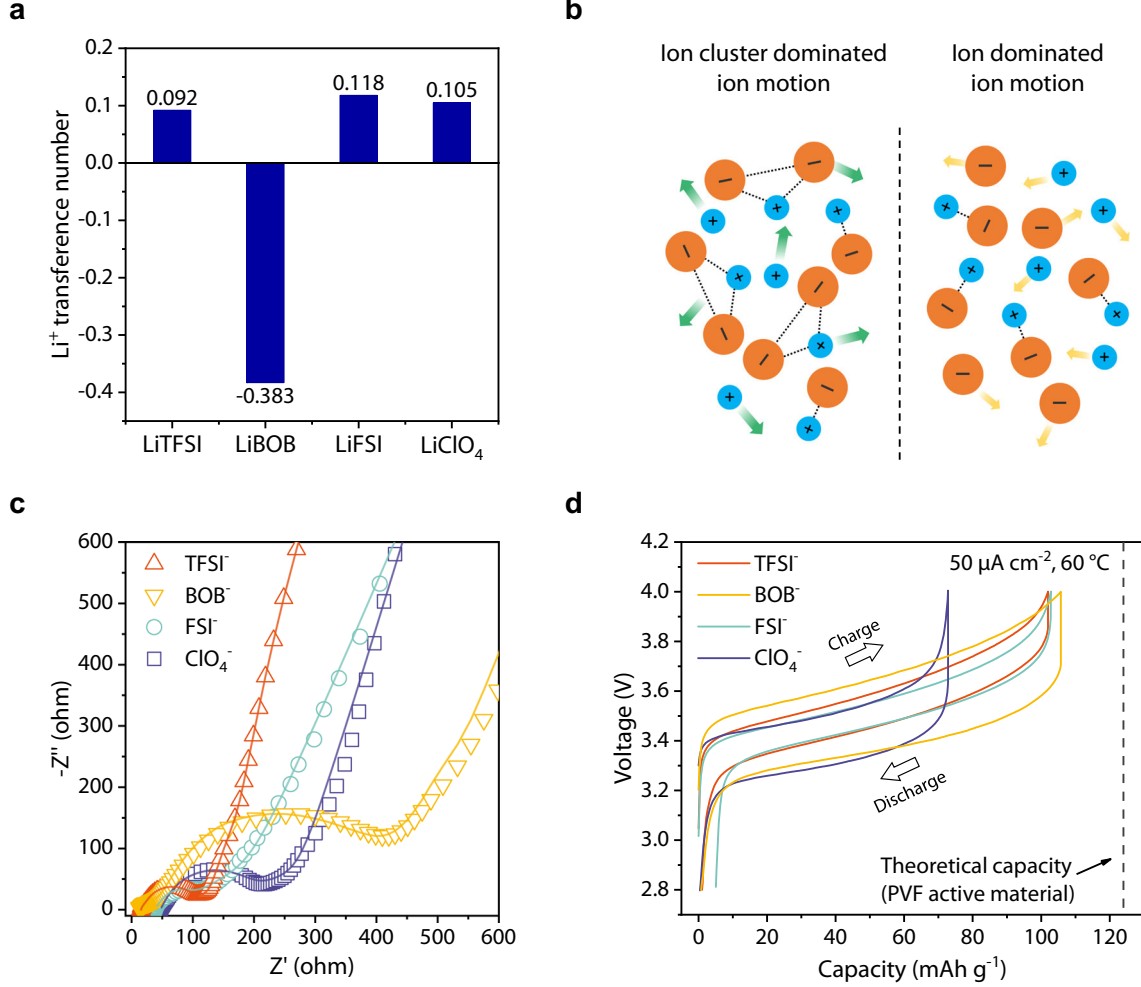

**Fig. 3 Effect of the salt anion on the ion aggregation and electrochemical energy storage performance. a** Lithium-ion transference number measured using steady-state current method. **b** Ion motion in SPEs dominated by aggregated ion cluster (left) and single ion (right). The dashed lines between ions represent short-range intracluster interactions and arrows correspond to the ion motion considered in SPEs. Considering ion cluster as noninteracting species provide an approximation more precise than the usual Nernst-Einstein equation to the $t_{Li+}$, able to explain the mechanism responsible for the negative value[48, 51]. **c** Nyquist plots of Li||PVF cells with different anion species before cycles. The solid line represents the impendence fitting results. **d** Charge-discharge curves of Li|SPE|PVF cells performed at 50 μA cm$^{-2}$, the theoretical capacity was marked as a dashed line. All the results were obtained at 60 °C.

hexafluorophosphate (LiPF$_6$) was used as salt in a carbonate-based non-aqueous electrolyte solution (Supplementary Fig. 14c, d), the voltage plateau shifts to ~3.27 V, compared to ~3.54 V plateau obtained with the LiTFSI-containing ether-based non-aqueous electrolyte solution. Similar differences were also presented by Kim et al. in anions impact of electrode reaction[56].

**Electrochemical energy storage performance of Li||PVF cells.**
Motivated by the cells' small polarization and discharge-capacity results with PEO-LiTFSI and PEO-LiBOB (Fig. 3d), we focused on these two SPEs for further studies of long cycling at 60 °C. After initial cathode activation (5 to 10 cycles), the capacities with PEO-LiBOB were measured to be 112, 104, and 107 mAh g$^{-1}$ at 20, 50, and 100 μA cm$^{-2}$, with good retention (Supplementary Fig. 15). However, as shown in Supplementary Fig. 16, the batteries show substantial polarization and capacity fade at higher current density (~67 mAh g$^{-1}$ at 300 μA cm$^{-2}$), which is as weak as ClO$_4^-$ and FSI$^-$ (Supplementary Fig. 17), despite the ionic conductance disparity among threes. The poor rate performance with LiBOB can be attributed to the formation of ions aggregations, as discussed in the previous section. The sluggish electrode

reaction deteriorates the cells performance at high current densities.

As shown in Supplementary Figs. 18, 19, the Li|PEO-LiTFSI|PVF cells enable good cycling stability and rate performances at 60 °C. Specifically, the initial capacity reaches 97 mAh g$^{-1}$ at 300 μA cm$^{-2}$ and maintains 90% after 1000 cycles (Fig. 2e). The electrochemical impedance spectroscopy measurements carried out during the charge-discharge process show that the ion transfer impedance of the cell increases after full charge (Supplementary Fig. 20). This increase can be attributed to the hindrance of the subsequent ion-pair binding due to the aggregation of anions inside the cathode during charging. As can be seen from the CVs of Supplementary Fig. 21, the charge storage mechanisms in Li|PEO-LiTFSI|PVF cells are mainly associated with surface-controlled processes. However, the electrode reaction is not the rate-determining step for most solid-state devices owing to the solid electrolytes' limited ion transport capability. Despite using 50 wt.% conductive agent to overcome PVF low electronic conductivity, the control experiment proves that the capacity contribution of the conductive agent at the tested voltage range is negligible (Supplementary

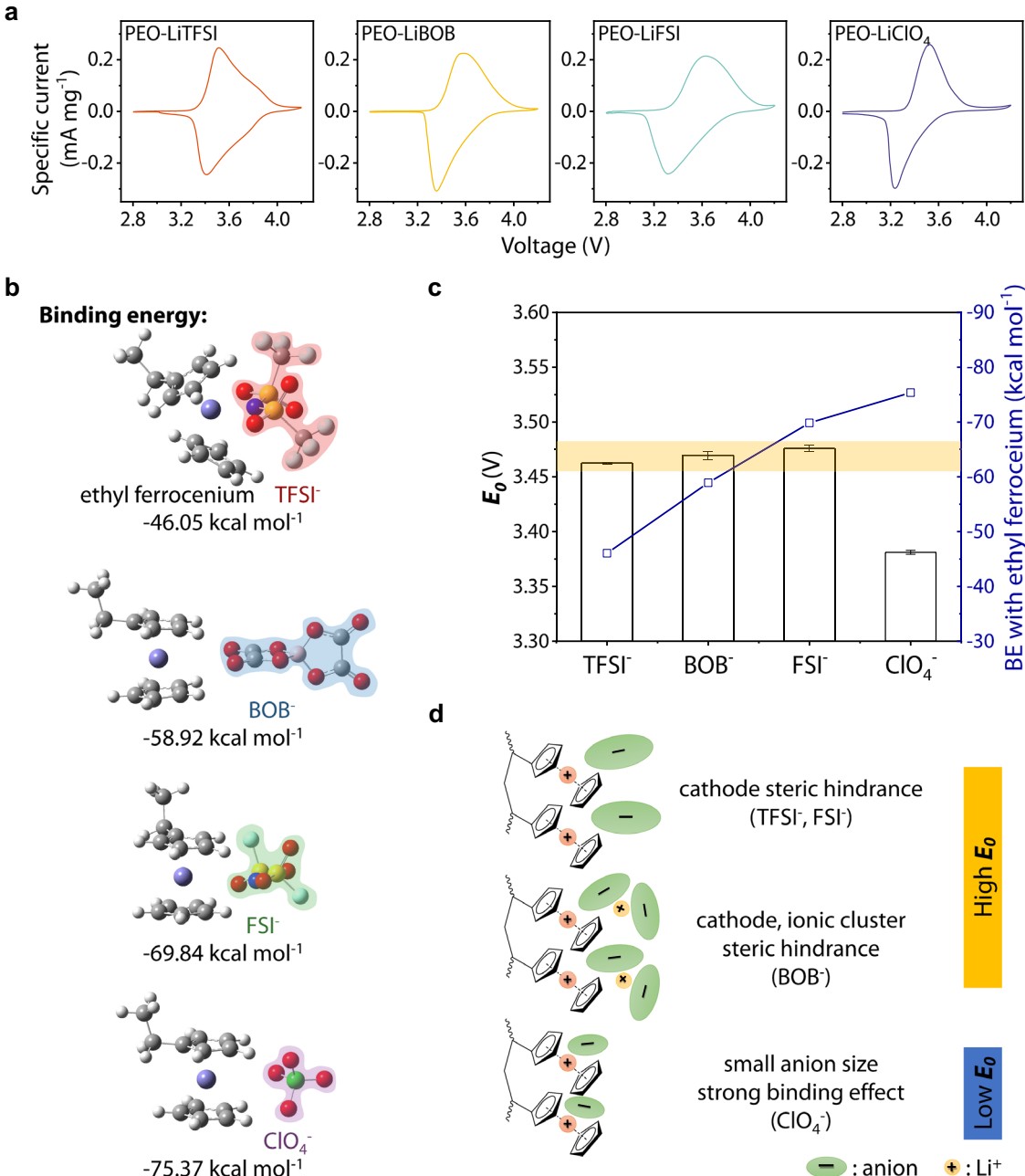

**Fig. 4 Effect of the salt anion on the PVF electrode reactions. a** Cyclic voltammetry curves of Li||PVF full cells constructed with different SPEs, performed at 0.2 mV s$^{-1}$, 60 °C. **b** The anions binding energy with ethyl ferrocenium. **c** The electrode potential ($E_O$) of SPEs, while $E_O$ is obtained by the mean of reduction and oxidation peaks in CVs (seen in Supplementary Table 5). Error bars represent the standard deviation calculated by the data sets. **d** The binding state of different anions to PVF.

Fig. 22). When the conductive agent decreased to 30 wt.%, Li||PVF cell exhibited discharge capacities of 100, 92, and 80 mAh g$^{-1}$ at currents of 50, 100, and 200 μA cm$^{-2}$, 60 °C, respectively (Supplementary Fig. 23). However, this does not affect the strategy's effectiveness proposed in this report. The addition of single-walled carbon nanotubes in the cathode can increase the proportion of active materials with the reduced polarization (Supplementary Fig. 24), which provides space for the anion-hosting electrode.

In Li||PVF cells, the anions and Li$^+$ migrate to the cathode and anode, respectively, and participate in the reaction during the charging process (Fig. 5a). This reduces the ion concentration on both electrodes surfaces, resulting in a relatively uniform distribution of salt concentration, avoiding the adverse consequences of concentration polarization (Supplementary Fig. 25). However, during the charging process of Li$^+$ acceptor electrodes (e.g., LiFePO$_4$), Li$^+$ is extracted from the cathode and deposited on the lithium metal. The number of ions in the SPE remains constant throughout the process. As a result, the concentration of Li$^+$ increases on the positive side and decreases on the negative side, leading a salt concentration gradient and an ion diffusion barrier (Supplementary Fig. 25). Concentration polarization could be detrimental in SPEs, accelerating anode degradation and cell failure (Fig. 5b).

To verify that the strategy could effectively enhance the lithium anode stability, we tested Li|PEO-LiTFSI|LiFePO$_4$ cells at 60 °C

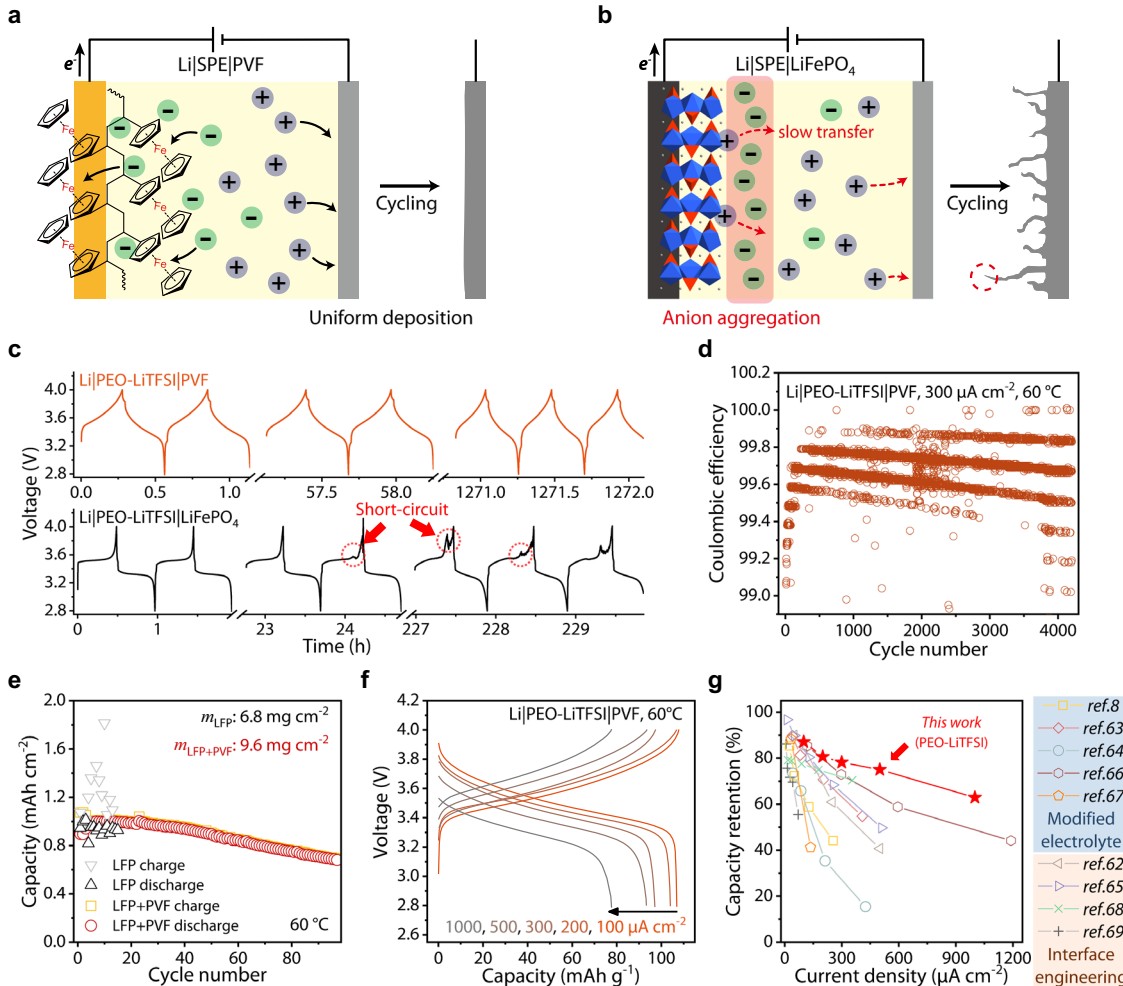

**Fig. 5 Electrochemical energy storage tests for lithium metal cells using the PEO-LiTFSI solid electrolyte.** Schematic diagram of anode morphology changes during cycling where (**a**) PVF and (**b**) LiFePO$_4$ served as cathode. **c** The potential profiles of Li||PVF (top) and Li||LiFePO$_4$ (bottom) cells at 300 μA cm$^{-2}$, 60 °C. **d** Coulombic efficiency of Li||PVF cell over 4000 cycles at 300 μA cm$^{-2}$, 60 °C. **e** Cycles with LiFePO$_4$ and mixed cathode (LiFePO$_4$:PVF = 1:1) at high areal loading. **f** The potential profiles of Li||PVF cells from 100 to 1000 μA cm$^{-2}$ and (**g**) the comparison between this work and the other reported advanced SPEs, classified in the graph by design strategy (detail seen in Supplementary Table 6).

(Supplementary Fig. 26). As shown in Fig. 5c, a sudden cell voltage loss of cycling occurred at the 24$^{th}$ hour at 300 μA cm$^{-2}$. The short circuit strongly affects the subsequent cycles, ultimately leading to a drop in terms of coulombic efficiency (Supplementary Fig. 26b). Previously published works correlated the cell performance degradation in the first few cycles with lithium deposition having unfavorable surface morphologies (e.g., dendrites)[57,58]. In contrast, the coulombic efficiency of the Li|PEO-LiTFSI|PVF cell maintained about 99.7% at least 4000 cycles at 60 °C, 300 μA cm$^{-2}$, with no sudden drop observed (Figs. 2e, 5d).

To further understand the anion-introduced electrode's performance, we run a couple of control experiments. The n-type organic electrode (Li$^+$-hosting) poly(anthraquinonyl sulfide) (PAQs) were set as cathode, and the results are similar to those of LiFePO$_4$, with short-circuit at 109$^{th}$ cycles (Supplementary Fig. 27). Besides, the performance of mixed cathode with PVF and LiFePO$_4$ proves the rationality of our design (Supplementary Fig. 28). Importantly, for the specific areal capacity around 1 mAh cm$^{-2}$ tests, positive electrodes with only LiFePO$_4$ as active material quickly developed a short circuit during the initial cycles (Fig. 5e, Supplementary Fig. 29) and failed to work. In contrast, mixed cathode (mass ratio, LiFePO$_4$:PVF = 1:1) showed clear cycling improvement, maintaining

more than 90 cycles (Fig. 5e). Zaghib et al. proved by in situ SEM that Li metal dendrites have higher hardness than pure lithium metal[59], which is the main reason for the failure of polymer solid-state batteries with conventional positive electrode active materials such as LiNi$_{0.6}$Mn$_{0.2}$Co$_{0.2}$O$_2$ and LiFePO$_4$[30,60,61]. In the ex situ postmortem electrodes and solid electrolyte cross-section SEM and EDX measurements disclosed in Supplementary Fig. 30, it can be noticed a smooth Li metal electrode surface after 50 cycles at 200 μA cm$^{-2}$ and 60 °C.

**Influence of the ionic conductivity of the battery performances.** Due to the weakness on the ionic conductivity, the unmodified PEO-LiTFSI usually performs poorly in the system with Li$^+$ as active charge carriers, especially its rate capability. The extended carriers enable appealing cell cycling performances, with capacities of 108, 107, 97, and 94 mAh g$^{-1}$ at current densities of 100, 200, 300, and 500 μA cm$^{-2}$, respectively (Fig. 5f). Even when the current increases to 1 mA cm$^{-2}$, the full cell still maintains 78 mAh g$^{-1}$ capacity. In Fig. 5g (numerical values in Supplementary Table 6), it is reported the comparison with other research works disclosed in the literature[10,62–69]. As can be seen, the strategy to couple PEO-LiTFSI solid electrolyte and PVF-

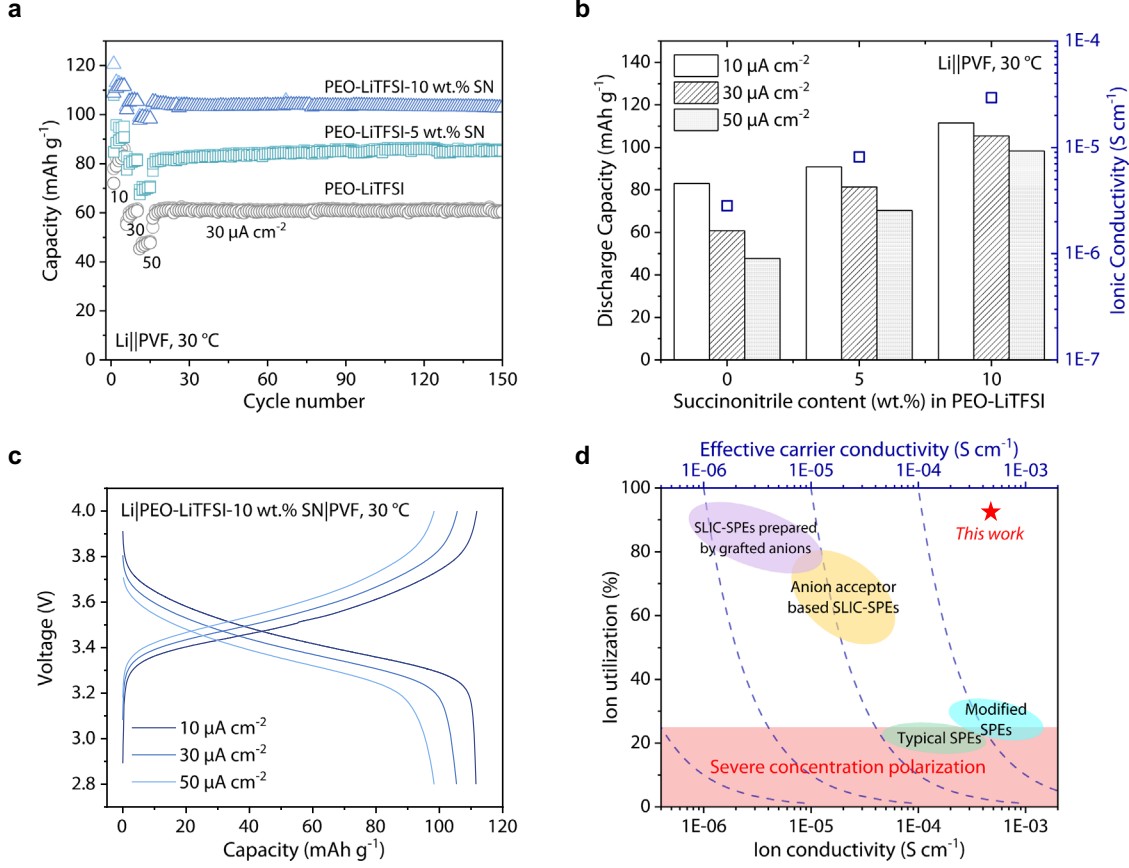

**Fig. 6 Li||PVF cells testing using succinonitrile as solid electrolyte additive. a** The cycle performance of SPEs with/without SN content. **b** Correlation between ionic conductivity of SPE and capacity of Li||PVF cell at 30 °C. **c** The potential profiles of Li|PEO-LiTFSI-10 wt.% SN|PVF cell at 30 °C. **d** Comparison of the effective carrier number of this work and other types of SPEs, where the ionic conductivity was collected at operating temperature.

based cathode allows a discharge capacity retention higher than 60% even at a current density of 1000 μA cm$^{-2}$ and 60 °C.

The improved current-carrying capacity motivates us to investigate the performance of Li||PVF cells when paired with SPEs with low ionic conductivity. PEO-LiTFSI exhibits $2.65 \times 10^{-6}$ S cm$^{-1}$ ion conductivity at 30 °C, far away from the ionic conductivity exhibited by the reported SPEs when tested (above $10^{-4}$ S cm$^{-1}$) (Supplementary Table 6). In this condition, Li||PVF cells exhibit capacities of 83, 60, 48 mAh g$^{-1}$ at 10, 30, and 50 μA cm$^{-2}$, 30 °C, respectively (Fig. 6a). As an electrolyte additive, succinonitrile (SN) could strengthen the segment movement ability of polymer, thereby enhancing SPEs' ionic conductivity[70,71]. After adding PEO-LiTFSI with 5, 10 wt.% SN, the ionic conductivity increased to $8.13 \times 10^{-6}$ and $2.93 \times 10^{-5}$ S cm$^{-1}$ at 30 °C (Fig. 6b, Supplementary Fig. 31), associated with the increase in capacity. Specifically, the cell with 10 wt.% SN exhibit capacity of 111, 105, 98 mAh g$^{-1}$ at 10, 30, and 50 μA cm$^{-2}$, 30 °C, respectively (Fig. 6c). Moreover, the Li||PVF cells with these SPEs show stable discharge capacity for more than 100 cycles at 30 μA cm$^{-2}$, 30 °C (Fig. 6a, Supplementary Fig. 32). Meanwhile, the CV curves exhibit stable and reversible redox behavior. Compared with the results of SN-free, the E$_0$ of electrode reaction is similar (Supplementary Fig. 33), proving that SN has no apparent influence on PVF redox. The anion-hosting cathode enables exploiting all the ions in the electrolyte as active charge carriers, thus unlocking appealing battery performances even with limited electrolyte ionic conductivity. The strategy of coupling ionic motion and electrode reaction allows high ion utilization (e.g., > 80 %) with a polymeric electrolyte having an ionic conductivity in the order of magnitude of $10^{-4}$ S cm$^{-1}$ (Fig. 6d)[6,7].

## Discussion

In summary, we developed polymer-based solid-state batteries by exploiting anions as effective carriers simultaneously with Li$^+$. The anion-hosting cathode PVF put the entire ion movement of SPEs into the electrochemical reactions, which produces an improved rate performance and promotes batteries operation at limited ionic conductance. In addition, the stable cycles of Li||PVF full cell prove that the anode deterioration effect, mainly contributed by concentration gradients, were circumvented effectively by reactive anion migration, which is essential for safe metal batteries. Besides, experiments and theoretical calculations clarified the effects of anion structure, binding energy and ion aggregation associated with the cell performance. Future investigations into high capacity and conductivity anion-hosting cathode are expected to follow. Since the migration and aggregation of anions differ from Li$^+$ in most SPEs, the disparity of reaction status between cathode and anode could present some complications and addressing this issue could be the subject of the future studies.

## Methods

**Synthesis of polyvinyl ferrocene and solid polymer electrolytes.** Polyvinyl ferrocene (PVF) was synthesized by free-radical polymerization. 2,2′-azoisobutyronitrile (AIBN, 99%, Aladdin) was dissolved in ethanol at 70 °C and filtered. After the cooling at room temperature of the filtered solution, the AIBN was recrystallized. In a typical PVF synthesis process, 3 g vinyl ferrocene (98%, Meryer) was dissolved in 5 ml dry toluene (H$_2$O, O$_2$ < 50 ppm, obtained from a VSPS-5 solvent purification system), and AIBN was used as the initiator. The mole ratio of monomer/ initiator = 100. The reaction was performed in a Schlenk tube (15 mL), with magnetic mixing last for 48 h under 60 °C. Add the obtained deep red solution to at least 200 ml of methanol (99%, Aladdin), filter to get precipitation, and wash 3

times in the Buchner funnel with methanol. The product is dry for 12 h under 60 °C to obtain yellow powder.

For solid polymer electrolytes synthesis, 0.6 g polyethylene oxide ($M_w = 6 \times 10^5$, Aladdin) and lithium salts (LiTFSI, 99%, LiFSI, 99%, LiClO$_4$, 99.9%, LiBOB, 98%, Aladdin) were dissolved in 15 mL acetonitrile (99%, Aladdin), under magnetic mixing over 24 h, and the mass of salts is determined by O:Li. Specifically, for the O:Li = 20:1 in SPEs, 195.7 mg LiTFSI or 132.1 mg LiBOB or 127.5 mg LiFSI or 72.5 mg LiClO$_4$ were added into PEO/acetonitrile slurry, and for the O:Li = 30:1 or 40:1 in PEO-LiBOB, 88.1 mg or 66.1 mg LiBOB were added. For the solid electrolyte with succinonitrile as additive (99%, Sigma-Aldrich), succinonitrile (30 mg for 5 wt.% SN and 60 mg for 10 wt.% SN) was added to acetonitrile with PEO and lithium salt. The obtained solution was coated on a round polytetrafluoroethylene plate ($d = 8$ cm, 0.5 cm depth) and dried over 72 h in an Ar-filled glovebox (H$_2$O, O$_2$ < 0.1 ppm), then further dried at 40 °C, 12 h under reduced pressure to obtain a self-supported membrane. Based on mass of PEO and plate diameter, SPEs thickness is 100 ± 5 µm. The electrolyte membrane was kept in an Ar-filled glovebox (H$_2$O, O$_2$ < 0.1 ppm) to prevent moisture contamination.

**Materials characterization**. The thermal gravimetric analysis was carried out with METTLER TOLEDO TGA/DSC$^{3+}$ at a temperature range of 30–800 °C under nitrogen atmosphere, with a heating rate of 10 °C min$^{-1}$. Fourier transform infrared (FT-IR, Bruker Tensor 27) were recorded between 400 and 4000 cm$^{-1}$. A field emission scanning electron microscope (FE-SEM, SU-6600) equipped with an energy dispersive spectrometer was used to characterize the samples' morphology. All the electrochemical characterization was performed using the electrochemical workstation (PARSTAT 1000 and CHI 660E). For the ex situ postmortem SEM and EDX cross-section measurements of the electrode and electrolyte samples, disassemble the cell in an Ar-filled glovebox (H$_2$O, O$_2$ < 0.1 ppm), and use a scalpel to cut the layer structure carefully, creating the cross-section. Place the cross -section on the pre-prepared sample table and keep it in the Ar atmosphere through plastic sealing. During the transfer to the SEM sample room, the sample was exposed to the air of no more than 20 s.

**Electrochemical measurements**. The conductive agent (Super P, 99%, average particle size ~ 40 nm), poly(vinylidene fluoride) (HSV900, 99.5%), LiFePO$_4$ (P198-S13, D50 = 1.302 µm, carbon-coated) and separator (Celgard 2325) were purchased from MTI China, dried at 80 °C at least 12 h before further used. The liquid electrolyte used were purchased from DodoChem (H$_2$O < 50 ppm). The electrode preparation of different active materials is carried out following the same method (PVF:conductive additive:binder = 4:5:1, LiFePO$_4$:conductive additive:binder = 7:2:1, mass ratio). The active material, Super P and poly(vinylidene fluoride) were dispersed in N-methyl pyrrolidone (99%, Aladdin) and ball milling for 4 h with 400 rpm to obtain a uniformly dispersed slurry, directly under air conditions. The ball milling was carried out in a cylindrical Teflon ball milling jar with an inner diameter of 3 cm and a depth of 7 cm, and agate ball milling beads with a diameter of 6 mm were used. For the electrode with high mass loading (LiFePO$_4$ and PVF + LiFePO$_4$), PEO were set as binder and dispersed in acetonitrile, the mass ratio of active materials:conductive:binder = 6:3:1 and PVF:LiFePO$_4$ = 1:1 in mixed cathode. The slurry was coated on aluminum foil (99.9%, thickness = 16 µm, with 1 µm conductive carbon coating, MTI China) using doctor blade, then dried under reduced pressure at 60 °C for 12 h. The electrodes were cut into disks ($d = 12$ mm) and stored in an Ar-filled glovebox (H$_2$O, O$_2$ < 0.1 ppm) for subsequent testing. Except where differently indicated, the mass loading of the active material was controlled at 1.0 mg cm$^{-2}$. The cells (coin type, CR2025) assembly was carried out in an Ar-filled glovebox (H$_2$O, O$_2$ < 0.1 ppm), with metallic lithium (99.9%, $d = 16$ mm, thickness = 200 µm, Canrd New Energy Technology) as counter electrode. Solid electrolyte membrane or separator (liquid electrolyte tested for Li||PVF cells) is placed between the positive and negative electrodes and assembled with the structure of conventional coin batteries.

The ionic conductivity of the electrolyte was measured by sandwiching the SPEs between two stainless steel electrodes (99.9%, $d = 16$ mm, thickness = 0.8 mm, Canrd New Energy Technology) and then recording the electrochemical impedance at open-circuit voltage in the potentiostatic mode over a frequency range of 10$^{-1}$–10$^5$ kHz using an alternating current perturbation of 10 mV, and 10 s open-circuit voltage applied before carrying out the EIS measurement. The following equation calculated ionic conductivity

$$\sigma = \frac{l}{RS}$$

$l$ is the thickness of the polymer electrolyte. $R$ is the bulk resistance of the polymer electrode, obtain from the intercept with the $x$-axis in EIS results. $S$ is the electrode area.

The lithium-ion transfer numbers of SPEs were measured by potentiostatic polarization to Li|SPEs|Li cells (coin type, CR2032), calculated from Bruce–Vincent–Evans equation

$$t_{Li^+} = \frac{I_{ss}(\Delta V - I_0 R_0)}{I_0(\Delta V - I_{ss} R_{ss})}$$

$\Delta V$ is the polarization voltage (10 mV), $I_0$ and $R_0$ are the initial current and interfacial resistance before polarization. $I_{ss}$ and $R_{ss}$ are the steady-state current and

interfacial resistance after polarization, respectively. The electrochemical impedance at open-circuit voltage in the potentiostatic mode over a frequency range of 10$^{-1}$–10$^5$ kHz using an alternating current perturbation of 10 mV.

The cell electrochemical impedance spectroscopy of the lithium metal coin cells was tested under an open circuit in the potentiostatic mode, with a frequency range of 10$^{-1}$–10$^5$ Hz and 10 mV potential perturbation applied. The cyclic voltammetry (CV) and linear sweep voltammetry (LSV) test were performed at a scan rate of 0.2 mV s$^{-1}$. Galvanostatic charge/discharge tests were performed within a voltage range of 2.8~4 V for PVF and LiFePO$_4$ on a LAND battery tester. Use a constant temperature cultivation incubator (30 °C) and an oven (60 °C) to control the temperature of the cells test, and the solid-state cells were kept at 70 °C at least 8 h before the formal test.

**DFT computational methods**. The optimization and frequency calculations were carried out on the B3LYP/6–311+G(d,p) level, and the optimized structures are free from imaginary frequency. The binding energies were calculated by $E_{binding} = E_{compound} - E_{anion} - E_{cation}$. The results are shown in Supplementary Table 5. All the above calculations were carried out with Gaussian 16 program package.

## Data availability
The data generated in main text are provided in the Source Data file.

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

## Acknowledgements

This research was supported in part by the National Natural Science Foundation of China (Nos. 51973171) [Ding], and Young Talent Support Plan of Xi'an Jiaotong University [Ding]. Natural Science Basic Research Program of Shaanxi (No. 2020-JC-09) [Ding]. Fundamental Research Funds for the Central Universities (xjh012020042) [Ding].

## Author contributions

The idea and project were conceived by Z.S. and S.D.; Z.S. designed the experiment. K.X. made important revisions to the full text and made key recommendations. J.C. performed polymer characterization; M.Y.L. performed DFT computational tests; Y.Q. performed the SEM tests; Y.Q.S. provided computational suggestion; A.A, Y.L. and Q.J. assisted in the result analysis and provided suggestions during the preparation of the manuscript; R.V.K. provides assistance in the analysis of electrochemical test results. All authors have given approval to the final version of the manuscript.

## Competing interests

The authors declare no competing interests.
