## [Peer Review File · Nature Communications]

Expanding the active charge carriers of solid polymer electrolytes in lithium-based battery using an anion-hosting cathodeREVIEWER COMMENTS

Reviewer #1 (Remarks to the Author):

A useful paper with good, reasonable and helpful data for R&D of solid polymer electrolytes. Though, the key points are reasonably concluded I would highly suggest not to underestimate recent literature with altered perspectives, as this work would be more objective, thus would have a higher impact. My comments/suggestions for a major revision are as follows:

1. "The safety requirement for lithium-ion batteries.....solid polymer electrolytes (SPEs) had been considered as a potential solution for constructing all-solid-state lithiums secondary batteries^{5,6}."

I understand, that polymers do not leak and have a higher flash point, but still they are highly combustible. Please make this safety statement more precise considering literature.

2. "One critical problem that has to be solved for constructing advanced SPEs is the movement of anions. The dissociation of lithium salt in SPEs depends on the dipolar interaction between the polymer chain (e.g., -CH₂-CH₂-O- in PEO) and Li⁺^{14,15}."

It is not simple a bipolar interaction, which is way to general. It is a chelating complexation. Literature suggestion: Cowie, J. M. G.; Cree, S. H., ELECTROLYTES DISSOLVED IN POLYMERS. Annu. Rev. Phys. Chem. 1989, 40, 85-113.

3. "In contrast, the non-coordinating anions contribute most of the ionic conductance in SPEs, meaning that the majority of ion migration is irrelevant to energy generation."

I highly agree. This is the reason why the Li⁺ diffusion coefficient is (significantly) more relevant. I suggest to modify and consider this recent literature:

<https://doi.org/10.1016/j.mattod.2020.11.025>

4. "This polarization results in critical performance degradation, such as increasing internal resistance and decreasing operating voltage²⁰, most importantly, promoting lithium metal dendrite growth²¹."

A good point. They may originate from these limitations. However, I think it necessary to clarify that the Li dendrites can have other origins, too (and are even observed in single ion conducting electrolytes, e.g. sulfides). Please revise for reasons of clarity considering the surface resistances as a main dendrite contributor, as well, also in LIBs, considering these references <http://dx.doi.org/10.1039/D0TA11775G> (for LIBs) and <https://doi.org/10.1016/j.isci.2020.101225> (for solid-state) and <https://doi.org/10.1016/j.matt.2020.03.015> (good summarizing review)

5. "To verify that minimizing the concentration polarization could effectively enhance the lithium anode stability, we tested LiFePO₄|PEO-LiTFSI|Li batteries under the same condition (Fig. S20). As shown in Fig. 5c, 5e, a micro short circuit occurred in LiFePO₄|Li at the 24th hour at 300 μ A cm⁻². The short circuit has not been repaired in subsequent cycles, leading to a continuous decayed in coulombic efficiency (Fig. S20b)."

Short circuits in cell with solid polymer electrolytes are extensively elaborated in recent works and should be not ignored in this passage: <http://dx.doi.org/10.1039/D1MA00009H> and <https://doi.org/10.1038/s41598-020-61373-9>

The former literature shows that a constant distance (realized by a simple spacer) can prevent short circuits originating from dendrites as the spacer prevents the shrinkage in electrode distance, thus the risk of short circuits. My suggestion in this regard (maybe for future work) is to validate these cells with such a spacer, as this would reveal more

systematic data (constant distance between the electrodes), as otherwise the PEO is mechanically prone to shrinkage.

Reviewer #2 (Remarks to the Author):

The article entitled "Expand the effective carriers in solid polymer electrolytes via anion-hosting cathode" by Sun et al. describes the use of polyvinyl ferrocene as positive active material in Li-metal polymer battery. Although some results are promising, the article needs a major revision before to be considered for publication in high impact journal like Nature Communications. Especially I have some concerns about cathode preparation and the loading used.

Here my comments:

1) As first in the introduction some missing articles about solid polymer electrolyte must be cited: a) Adv. Mater. 2018, 30, 1704436; b) Materials 12 (23), 2019, 3892; c) Nano Energy 86 (2021) 106057; d) J. Mater. Chem. A, 2021, 9, 11794-11801; e) Journal of Power Sources 450 (2020) 227614

2) The authors must add a SEM image and EDS mapping of the cathode and include the results in Figure 2.

3) The authors wrote: "The same method was used to prepare LiFePO₄ electrodes with a ratio of 7:2:1 and the Super P electrode in a ratio of 85:15 (conductive agent: binder).": there is not polymer SPE at all at the cathode? What did they use? Just a drop of liquid electrolyte? If they use just liquid electrolyte instead of SPE at the positive electrode the performance are not impressive. The material loading of 1 mg/cm² is too low: the value is not representative for any future application (too low energy density). The authors should test capacity at least of 1.0-1.5 mAh/cm² (ideally > 2mAh/cm²) and compare the performance results with what it's present in the manuscript (1mg/cm² active material). The authors must do the same for LiFePO₄-Li metal battery. Cathode preparation is the most critical point of the work.

4) Have other additives been tested? What about TEGDME? Propylene carbonate? Succinonitrile as plastic crystals is solid at room temperature. Probably by using other additives the performances may improve

5) I am not totally convinced about the proposed lithium conductivity mechanism. The authors must refer the following previous works: a) Current Opinion in Electrochemistry 9, 2018, 56-63; b) ACS Appl. Mater. Interfaces 2018, 10, 28; c) Chemical Engineering Journal 394 (2020) 1248472

6) What is thickness of lithium metal? High energy dense lithium metal battery requires thin or free lithium. Considering the high homogenous lithium plating reported in the manuscript, the authors should prepare a battery with a lithium less configuration cell (direct plating on copper foil).

7) I find quite strange that LiTFSI behaves better than LiFSi. Is there any influence of Fluorine on the performances? Moreover the authors should cite some previous work about Li-salt contribution to SPE performance (a) Materials Science and Engineering: R: Reports 134, 2018, 1-21; b) Angew.Chem. Int. Ed. 2019, 58,15978 -16000)

8) Does a mix NMC-PVF improve NMC stability with PEO electrolyte? PVF could be used as buffer for other cathodes (NMC,LFP, etc etc)

9) What is the possible side reactions related to capacity fading? I believe that these reactions will be much more evident at higher loadings.

10) About Li-metal deposition: the authors should add cross section SEM image and EDS mapping of Li-SPE interface (PVF vs LFP batteries) and cite the following previous works adding a discussion(a) <https://doi.org/10.1016/j.matt.2020.03.015>; b) Nano letters 18 (12), 2018, 7583-7589; c) Nano letters 20 (3), 2019, 1607-1611; d) Communications Chemistry 2, 2019, 131)

11) What are other possible polymers with comparable properties? The authors must cite some previous work (a) Chemical Engineering Journal 416 (2021) 129171; b) Materials 12 (11), 2019, 1770) Did authors consider the possibility to use other metallocene (cobaltocene)?

Reviewer #3 (Remarks to the Author):

Authors investigate a solid-state dual-ion battery using the typical PEO as polymer matrix and a redox-active polymer as cathode. This system is introduced to avoid the concentration polarization observed with SPEs and provide an active use of the anions, which is significant to the field.

The whole paper is focused on the fact that the improved electrochemical performance of PVF|PEO:LiTFSI|Li is due to the avoidance of concentration gradients. However, there is not enough proof to confirm the hypothesis and therefore further experiments and explanations are needed. This is justified in the following specific as well as general comments:

- 1. I suggest the authors to mention that this is a dual-ion battery as it is already in use term to describe this type of battery where anions take place in the reaction. This might help the reader understand the topic.**
- 2. Page 3 line 55, what do authors mean by ionizing less free Li+? Is it that more ion clusters and aggregates are prone to form? I suggest to clarify**
- 3. Page 3 line 55, authors mention that the cost of using SISPEs is the low ionic conductivity due to the lack of solvation ability. The ionic conductivity of SISPEs is considered low compared to conventional SPE due to the fact that anions do not contribute to the ionic conductivity, so it is not a fair comparison. In addition, they ascribed it to the lack of solvation ability. The solvation ability is given by the polymer matrix either if it is in a conventional SPE or SISPEs. Further clarification would be beneficial.**
- 4. Page 5, authors should also provide information about how the Tg changes with the different salts.**
- 5. Page 5 line 97. Authors should provide a reference of the method used to calculate the transference number.**
- 6. Page 5 line 106. Authors attribute the high charge transfer resistance with PEO-LiBOB to the negatively charged clusters. Why? What about contribution from the different interphases? EIS does not prove that the high Rct is from clusters. Authors should provide other more accurate proof of that.**
- 7. Page 5 line 107. "Low tLi+ operate poorly" is not correct. PEO, despite its low tLi+, has overall good performance (at least at high temperatures) confirmed by multiple articles and commercial batteries built with this system. I suggest to rephrase it.**
- 8. Page 5 line 110. While the ionic conductivity of the SPE will affect the overpotential of the batteries, there are other factors that affect as well. And if the overpotential is controlled by the ion conductivity why do the authors study the anion impact?**
- 9. Page 6 line 138. Authors use DFT to model the interaction between ferrocenium and anions but then provide statements of what happens in the polymer system: "the folded long-chain polyvinyl ferrocenium suppress the combination of large anions, weakening the effect of ion-pairing on the electrode potential." However, there is no proof of such statement. This section would benefit from further explanations and mentioning more clearly to which experimental results they are referring to. While the difference in experimental electrode potentials is large, the difference from the calculations is very small. In addition, what is the theoretical value? Is the anion the only contributor to the shift in voltage plateau? But before this difference in redox peaks and plateaus was assigned to the difference in ionic conductivity.**
- 10. Page 6 line 143. The stability of the electrode was examined with LiPF6, as the anion has a large impact in the electrochemistry, LiTFSI would be a better choice for comparison.**
- 11. Page 7 line 157. The poor rate performance of LiBOB is assigned to the formation of**

ions aggregations due to the negative t^+ . However, similar poor performance is observed for LiClO_4 and LiFSI despite them having similar t^+ as LiTFSI and LiFSI having similar ionic conductivity as LiTFSI . Why? Maybe using a different polymer matrix with higher t^+ or different anion mobility would provide better insight on the effect on anion and its mobility.

12. Page 8. Authors compare PVF with LFP electrodes. However, they have two completely different redox chemistry and surface chemistry. Maybe another redox active polymer that is not p-type would be a better choice of comparison. And maybe further comparison with other p-type polymers would also benefit this study.

13. Page 8. Authors study the effect of ionic conductivity. By adding plasticizers, the ionic conductivity improves and so does the battery performance. This would suggest that the reason behind the improved performance is the ionic conductivity but do not give insight on the effect of the anion or the reaction of the PVF.

14. Page 8 line 215. Authors provide the CV of the battery with SN additive. It would be interesting to show the CV with the same conditions (temperature and scan rate) without SN in order to compare both systems. With SN the difference between the oxidation and reduction peaks is large, how does that compare to the system without SN? If it solely depends on the anion as previously discussed it should be the same with or without SN. This could provide additional proof if the improvement is coming from the higher ionic conductivity or the anion.

In general, further understanding and investigation of the anion migration should be carried out in order to confirm the hypothesis of this paper. I also suggest the authors to check other papers where they analyze the anion mobility in different systems. LiTFSI is the best performing salt for solid polymer electrolytes for most of the reported systems with different polymer matrices and also cathode materials, therefore if authors want to investigate the effect of the anion migration other polymer systems should be included and not only vary the salt.

Response to Reviewers' Comments

Reviewer #1 (Remarks to the Author):

A useful paper with good, reasonable and helpful data for R&D of solid polymer electrolytes. Though, the key points are reasonably concluded I would highly suggest not to underestimate recent literature with altered perspectives, as this work would be more objective, thus would have a higher impact. My comments/suggestions for a major revision are as follows:

Response:

We appreciate the reviewer's valuable comments very much. We have considered the reviewer's comments and revised the manuscript accordingly. Concerning the reviewer comment about recent literature, we have added several new references at different parts of the paper. We believe we have covered a great part of the literature.

1. *"The safety requirement for lithium-ion batteries.....solid polymer electrolytes (SPEs) had been considered as a potential solution for constructing all-solid-state lithium secondary batteries^{5,6}."*

I understand, that polymers do not leak and have a higher flash point, but still they are highly combustible. Please make this safety statement more precise considering literature.

Response:

We agree with the reviewer's point of view. Although SPE has higher flashing points, it might still represent some problems due to its combustibility. There are several attempts in the literature to address this point, such as careful structural design, adding flame-retardant components or building a non-flammable main chain (polyionic liquid or Fluorine-rich et al.). Therefore, we have made corresponding supplementary clarifications in the revised version to make this statement more accurate.

Since Wright and coworkers reported the ionic conductivity of the mixture of

polyethylene oxide (PEO) and alkali metal salts⁵, solid polymer electrolytes (SPEs) are attractive for constructing all-solid-state lithium secondary batteries due to the lack of leakage risk⁶. Recently works have addressed the flammability problem of such SPEs using different strategies^{7,8}.

Ref:

7. Wan, J., Xie, J., Kong, X., Liu, Z., Liu, K., Shi, F. & Cui, Y. Ultrathin, flexible, solid polymer composite electrolyte enabled with aligned nanoporous host for lithium batteries. *Nat. Nanotech.* **14**, 705-711 (2019).
8. Choi, Y. G., Shin, J. C., Park, A., Jeon, Y. M., Kim, J. I., Kim, S. & Park, J. H. Pyrrolidinium- PEG Ionic Copolyester: Li- ion accelerator in polymer network solid- state electrolytes. *Adv. Energy Mater.* **11**, 2102660 (2021).

2. “One critical problem that has to be solved for constructing advanced SPEs is the movement of anions. The dissociation of lithium salt in SPEs depends on the dipolar interaction between the polymer chain (e.g., $-CH_2-CH_2-O-$ in PEO) and Li^+ ^{14,15}.”

It is not simple a bipolar interaction, which is way to general. It is a chelating complexation. Literature suggestion:

Cowie, J. M. G.; Cree, S. H., ELECTROLYTES DISSOLVED IN POLYMERS. *Annu. Rev. Phys. Chem.* 1989, 40, 85-113.

Response:

We are very grateful to the reviewers for his/her valuable suggestion. We agree that dipole interaction is very general and cannot reflect the complexity of the process. Therefore, we have made corresponding changes in the discussion part, referring to the revision's supplement literature.

In PEO, the repeat unit (CH_2-CH_2-O) show chelating complexation with Li^+ ¹⁷. Specifically, the PEO chains can adopt a helical conformation that presents the optimum distances for O-Li interactions similar to the crown structures^{18,19}.

Ref:

18. Cowie, J. M. G. & Cree, S. H. Electrolytes dissolved in polymers. *Annu. Rev. Phys. Chem.* **40**, 85-113 (1989).

3. “In contrast, the non-coordinating anions contribute most of the ionic conductance in SPEs, meaning that the majority of ion migration is irrelevant to energy

generation.”

I highly agree. This is the reason why the Li^+ diffusion coefficient is (significantly) more relevant. I suggest to modify and consider this recent literature:

<https://doi.org/10.1016/j.mattod.2020.11.025>

Response:

We thank the reviewer for the suggestion. We have checked the suggested paper and other recent literature on ion diffusion in SPEs and made additional comments in the revision.

The facile movement of the anions and the scarce supply of cations facilitates the uneven deposition of Li onto surface protrusions at high current density, leading to the self-amplification process of dendritic growth²²⁻²⁴.

Ref:

23. Stolz, L., Homann, G., Winter, M. & Kasnatscheew, J. The Sand equation and its enormous practical relevance for solid-state lithium metal batteries. *Mater. Today* **44**, 9-14 (2021).

4. “*This polarization results in critical performance degradation, such as increasing internal resistance and decreasing operating voltage²⁰, most importantly, promoting lithium metal dendrite growth²¹.*”

A good point. They may originate from these limitations. However, I think it necessary to clarify that the Li dendrites can have other origins, too (and are even observed in single ion conducting electrolytes, e.g. sulfides). Please revise for reasons of clarity considering the surface resistances as a main dendrite contributor, as well, also in LIBs, considering these references

<http://dx.doi.org/10.1039/D0TA11775G> (for LIBs)

<https://doi.org/10.1016/j.isci.2020.101225> (for solid-state)

<https://doi.org/10.1016/j.matt.2020.03.015> (good summarizing review)

Response:

We agree with the reviewers' comments and consider all the suggested reasons in the recommended papers. As highlighted in the recommended literature, the ionic conductivity, concentration polarization, and mechanical properties can cause

dendrites in SPEs. Therefore, our discussion of dendrite's forming mechanisms in the original manuscript has been revised, and more details have been added. In the revised version, we refer to the works about multi-systems previous reported and modify the discussion in this part.

The polarization results in critical performance degradation, such as increasing internal resistance and decreasing operating voltage²⁷. Coupling with the low Li⁺ migration increasing the electrode/electrolyte interface resistances significantly to accelerate the deterioration of anodes²⁸⁻³⁰.

Ref:

28. Klein, S., Bärmann, P., Fromm, O., Borzutzki, K., Reiter, J., Fan, Q. & Kasnatscheew, J. Prospects and limitations of single-crystal cathode materials to overcome cross-talk phenomena in high-voltage lithium ion cells. *J. Mater. Chem. A* **9**, 7546-7555 (2021).
29. Homann, G., Stolz, L., Winter, M. & Kasnatscheew, J. Elimination of “voltage noise” of poly (ethylene oxide)-based solid electrolytes in high-voltage lithium batteries: linear versus network polymers. *iScience* **23**, 101225 (2020).
30. Cao, D., Sun, X., Li, Q., Natan, A., Xiang, P. & Zhu, H. Lithium dendrite in all-solid-state batteries: growth mechanisms, suppression strategies, and characterizations. *Matter* **3**, 57-94 (2020).

5. “To verify that minimizing the concentration polarization could effectively enhance the lithium anode stability, we tested LiFePO₄|PEO-LiTFSI|Li batteries under the same condition (Fig. S20). As shown in Fig. 5c, 5e, a micro short circuit occurred in LiFePO₄|Li at the 24th hour at 300 μA cm⁻². The short circuit has not been repaired in subsequent cycles, leading to a continuous decayed in coulombic efficiency (Fig. S20b).”

Short circuits in cell with solid polymer electrolytes are extensively elaborated in recent works and should be not ignored in this passage:

<http://dx.doi.org/10.1039/D1MA00009H>

<https://doi.org/10.1038/s41598-020-61373-9>

Response:

The reviewer's literature suggestion for this point is very reasonable. In addition, there have been related in-depth studies on the short circuit phenomenon of

SPE-based solid-state batteries. Thus, we refer to the recommended studies and review the latest related work. Additional discussion and citations have been added to the revised version.

A similar degradation mechanism has been intensively studied in recent works. It was confirmed that high current density and lithium metal deposition accelerates the initial micro short circuit phenomenon, even in the first few cycles^{56,57}.

Ref:

56. Stolz, L., Homann, G., Winter, M. & Kasnatscheew, J. Realizing poly (ethylene oxide) as a polymer for solid electrolytes in high voltage lithium batteries via simple modification of the cell setup. *Mater. Adv.* **2**, 3251-3256 (2021).
57. Homann, G., Stolz, L., Nair, J., Laskovic, I. C., Winter, M. & Kasnatscheew, J. Poly (Ethylene oxide)-based electrolyte for solid-state-lithium-batteries with high voltage positive electrodes: evaluating the role of electrolyte oxidation in rapid cell failure. *Sci. Rep.* **10**, 1-9 (2020).

The former literature shows that a constant distance (realized by a simple spacer) can prevent short circuits originating from dendrites as the spacer prevents the shrinkage in electrode distance, thus the risk of short circuits. My suggestion in this regard (maybe for future work) is to validate these cells with such a spacer, as this would reveal more systematic data (constant distance between the electrodes), as otherwise the PEO is mechanically prone to shrinkage.

Response:

Thank you very much for the valuable suggestions of the reviewers. In the main part of this work, the carriers' expansion approaches allowed the polymer electrolyte to sustain more than 4000 cycles without a short circuit. Most of the similar electrolytes in the literature could not last more than 30 cycles. In addition, we have added to the revised manuscript the results of high mass loading tests (**Fig. S30, S32**). Due to the large volumetric changes at high loading, more pronounced deformation is taking place in the anode and cathode during the cycles. This leads to the failure of the batteries with LFP cathodes after a few cycles, while our designed cathode with PVF addition could survive cycling for more than 100 cycles at such a high salt loading. However, we also agree with the reviewer's suggestion that controlling the

anode-cathode spacing can limit the growth of dendrites and believe that this strategy will be of great help for mixed or high-load electrodes in future research.

Reviewer #2 (Remarks to the Author):

The article entitled “Expand the effective carriers in solid polymer electrolytes via anion-hosting cathode” by Sun et al. describes the use of polyvinyl ferrocene as positive active material in Li-metal polymer battery. Although some results are promising, the article needs a major revision before to be considered for publication in high impact journal like Nature Communications. Especially I have some concerns about cathode preparation and the loading used.

Response:

We thank the reviewer for the careful review of our manuscript and the insightful comments. We respond to each question of the reviewer one by one below. We have considered all the reviewer's comments and modified our paper accordingly. Overall, the paper has been subjected to a major revision with lots of discussion, experimental results and simulation results have been added to support our claims. The revised texts in the manuscript and supporting information are highlighted in yellow. Please find the details below.

1) As first in the introduction some missing articles about solid polymer electrolyte must be cited:

- a) Adv. Mater. 2018, 30, 1704436;
- b) Materials 12 (23), 2019, 3892;
- c) Nano Energy 86 (2021) 106057;
- d) J. Mater. Chem. A, 2021, 9, 11794-11801;
- e) Journal of Power Sources 450 (2020) 227614

Response:

We are very grateful to the reviewers for their suggestions. We have added the recommended references and comments of related work in the revised edition. The introduction part now provides a more comprehensive review of the reported work.

The safety requirement for lithium-ion batteries and the demand for lithium metal anode has prompted researchers to look for solid-state alternatives to liquid organic

electrolytes¹⁻⁴.

...

Typical strategies include designing the polymer segment structure¹⁰⁻¹² and combining polymer matrix with ceramic^{13,14} or inorganic solid electrolytes^{15,16}.

...

The facile movement of the anions and the scarce supply of cations facilitates the uneven deposition of Li onto surface protrusions at high current density, leading to the self-amplification process of dendritic growth²²⁻²⁴.

Ref:

4. Mauger, A., Julien, C. M., Paoella, A., Armand, M. & Zaghbi, K. Building better batteries in the solid state: A review. *Materials* **12**, 3892 (2019).
11. Fu, C., Iacob, M., Sheima, Y., Battaglia, C., Duchêne, L., Seidl, L. & Remhof, A. A highly elastic polysiloxane-based polymer electrolyte for all-solid-state lithium metal batteries. *J. Mater. Chem. A* **9**, 11794-11801 (2021).
12. Zhao, Y., Bai, Y., Liu, A., Li, W., An, M., Bai, Y. & Chen, G. Polymer electrolyte with dual functional groups designed via theoretical calculation for all-solid-state lithium batteries. *J. Power Sources* **450**, 227614, (2020).
14. Wang, Z., Tan, R., Wang, H., Yang, L., Hu, J., Chen, H. & Pan, F. A Metal-organic- framework- based electrolyte with nanowetted interfaces for high- energy- density solid- state lithium battery. *Adv. Mater.* **30**, 1704436 (2018).
24. Yuan, C., Gao, X., Jia, Y., Zhang, W., Wu, Q. & Xu, J. Coupled crack propagation and dendrite growth in solid electrolyte of all-solid-state battery. *Nano Energy* **86**, 106057 (2021).

2) The authors must add a SEM image and EDS mapping of the cathode and include the results in Figure 2.

Response:

The reviewer's suggestion on this issue is very reasonable. We agree that electrode morphology and element distribution is important. Therefore, the SEM and EDS mapping of the electrode at different magnifications. The results prove that the active and conductive materials are evenly mixed without significant agglomeration.

Fig. S2. The SEM and EDS mapping of cathode

3) The authors wrote: “The same method was used to prepare LiFePO_4 electrodes with a ratio of 7:2:1 and the Super P electrode in a ratio of 85:15 (conductive agent: binder).”: there is not polymer SPE at all at the cathode? What did they use? Just a drop of liquid electrolyte? If they use just liquid electrolyte instead of SPE at the positive electrode the performance are not impressive. The material loading of 1 mg/cm^2 is too low: the value is not representative for any future application (too low energy density). The authors should test capacity at least of $1.0\text{-}1.5 \text{ mAh/cm}^2$ (ideally $> 2\text{mAh/cm}^2$) and compare the performance results with what it’s present in the manuscript (1mg/cm^2 active material). The authors must do the same for $\text{LiFePO}_4\text{-Li}$ metal battery. Cathode preparation is the most critical point of the work.

Response:

We thank the reviewer for raising this point. We totally agree with the reviewer that the preparation and loading of the electrode are crucial for the current work. PVDF was used as the binder in the electrode preparation part without any liquid addition on the electrode surface. Since SPE-based solid-state batteries are usually tested under high-temperature (this work is the same), near the melting point of SPEs, thus, a part of SPE can penetrate the cathode and provide ion transmission. In previous works, the classical binder (PVDF) has been proved to maintain the operation of the solid-state battery (*Adv. Mater.* **2020**, *32*, 2000399, *Nano Energy* **2018**, *46*, 176, *Angew. Chem. Int. Ed.* **2021**, *60*, 12931). Besides, for the battery test at room temperature, we kept it at $70 \text{ }^\circ\text{C}$ for at least 8 hours before the formal test. We added these operation details in the experimental part.

We agree with the reviewer's concern about electrode mass loading and have performed new experiments with different mass loading. The current carrying capacity of solid electrolytes (especially SPEs) is much lower than that of liquid electrolytes, which causes the load capacity of the electrode to be incomparable with liquid-based batteries and usually cannot achieve high areal capacity. On the other hand, the PVF used in this work is a typical organic material, and the inherent low ion/electronic conductivity requires a large amount of conductive agent for sufficient electrochemical reactions even in liquid systems (*Energy Environ. Sci.* **2013**, *6*, 2280, *Nat. Rev. Chem.* **2020**, *4*, 127). Therefore, the high loading is a massive challenge for PVF electrodes.

Inspired by the reviewer's suggestion of mixing PVF with the traditional cathode (*comments 8*), we performed the cycling of physically mixed PVF and LiFePO₄ and compared them with LFP alone at high mass loading. In addition to the advantages of anion-extended carriers, the mixed electrode can provide an ideal areal capacity. Even if PVF only occupies 50% of the active material, its cycle stability is still much better than that of the LFP electrode. We believe that if the electrodes are all anion acceptor materials with high conductivity, the cycle life can be improved more significantly. In addition, the modification of the surface structure of the traditional lithium-ion acceptor cathode material or the coating of the anion acceptor material also has great potential.

Fig. S30. High mass loading performance. The initial area capacity was controlled around 1 mAh cm⁻². (a) Cycle stability and (b), (c) charge/discharge curves of different cathode. Benefit from anion-extended carriers, the mixed electrode can provide an ideal areal capacity and stable cycles, even if PVF only occupies 50% of the active material.

4) Have other additives been tested? What about TEGDME? Propylene carbonate? Succinonitrile as plastic crystals is solid at room temperature. Probably by using other additives the performances may improve

Response:

We thank the referee for the suggestions. Regarding the reviewer's comment about succinonitrile, we would like to emphasize that we used it to explore the batteries operation at low ionic conductivity. The conductivity of PEO-LiTFSI and PEO-LiTFSI-5wt% SN are 2.82×10^{-6} and 8.13×10^{-6} S cm⁻¹ at room temperature, much lower than the reported other SPEs. The results proved that the cell can still achieve reasonable stability despite the low conductivity. When 10 wt% SN were added, higher conductivity was obtained 2.93×10^{-5} S cm⁻¹. While this value is still lower than other tested SPEs ($\sim 10^{-4}$ S cm⁻¹), the battery performance significantly improved.

We very much agree with the suggestions for constructing better performance

electrolytes. TEGDME and PC are common liquid components with high boiling points, which can greatly improve the ionic conductivity of the modified SPEs. However, our concern is that the addition of liquid components will shake the definition of SPEs, which is our focus in this paper. Even with a small amount of liquid additives, some researchers still question the ion transport mechanism (*Adv. Mater.* **2020**, *32*, 1907375), pointing out its transport mode is similar to gel polymer electrolytes (GPEs). Therefore, we chose succinonitrile as the plasticizer to keep the paper's focus on solid-state batteries. Also, succinonitrile improves the capability of polymer segment with less impact on other properties of SPEs, which allow us to investigate other variables.

Fig. S33. AC impedance of PEO-LiTFSI with 5wt% and 10wt% SN at 30 °C.

Fig. R1. Cycle and capacity of PVF|Li batteries with/without SN content at 30 °C

5) I am not totally convinced about the proposed lithium conductivity mechanism. The authors must refer the following previous works:

- a) *Current Opinion in Electrochemistry* 9, 2018, 56-63;
- b) *ACS Appl. Mater. Interfaces* 2018, 10, 28;
- c) *Chemical Engineering Journal* 394 (2020) 1248472

Response:

We are very grateful for the suggestions made by the reviewers, which we found very helpful to improve the quality of our paper. Based on the reviewer's suggestion, we have modified the discussion about the Li^+ movement mechanism in the revision.

Yet, the weak polymer chain motion leads to insufficient ion transport for SPEs even at high temperature^{3,9}.

...

In PEO, the repeat unit ($\text{CH}_2\text{-CH}_2\text{-O}$) show chelating complexation with Li^+ ¹⁷. Specifically, the PEO chains can adopt a helical conformation that presents the optimum distances for O-Li interactions similar to the crown structures^{18,19}.

Ref:

- 9. Commarieu, B., Paoletta, A., Daigle, J. C. & Zaghbi, K. Toward high lithium conduction in solid polymer and polymer-ceramic batteries. *Curr. Opin. Electrochem.* 9, 56-63 (2018).
- 17. Zhao, Y., Bai, Y., Li, W., Liu, A., An, M., Bai, Y. & Chen, G. Semi closed coordination structure polymer electrolyte combined in situ interface engineering for lithium batteries. *Chem. Eng. J.* 394, 124847 (2020).

6) What is thickness of lithium metal? High energy dense lithium metal battery requires thin or free lithium. Considering the high homogenous lithium plating reported in the manuscript, the authors should prepare a battery with a lithium less configuration cell (direct plating on copper foil).

Response:

Thanks for the suggestion raised by the reviewer. The thickness of lithium foil is about 200 μm . We agree with the reviewer's view that high energy density batteries require less lithium or no lithium electrodes. Mortified by the excellent anode lithium stability observed in the current, we have carried out tests on lithium-free anode batteries. However, the performance was not so promising. The results indicated some reaction between the solid electrolyte and the anode, despite the stable deposition of

lithium. We will show the results of the experiment and conduct an in-depth analysis:

The dual-salts system has shown great potential in studying metal-anode in liquid batteries (*Nano Lett.* **2021**, *21*, 3310, *Nat. Energy* **2020**, *5*, 693, *Nat. Energy* **2019**, *4*, 683). Its unique solvation structure can achieve high coulombic efficiency of Li-Cu batteries and improve the cycle life. We tested our designed SPEs, with single/dual salts, in Li-Cu batteries. Due to the low transport of Li^+ , the LiTFSI single-salt system shows short circuits in the first cycle (**Fig. R2**). On the other side, the Li-Cu battery with PEO-LiTFSI-LiBOB can achieve depositing/plating successfully even at higher current density due to the solvation structure regulation. However, its coulombic efficiency was about 80% under 0.2 mA cm^{-2} , 0.2 mAh cm^{-2} , which is unacceptable for lithium-free batteries (**Fig. R3**).

Fig. R2. The performance of Li-Cu batteries with single/dual salts. The O: Li was controlled at 20: 1 and the mole ratio in the dual salts system is 1: 1. All tests were conducted at $60 \text{ }^\circ\text{C}$.

We believe that the stable lithium electroplating/stripping with low coulombic efficiency are caused by the continuous marginal reaction of lithium metal newly deposited on Cu with the electrolyte. Moreover, even if much well-designed liquid system can show high coulombic efficiency in the Li-Cu system (*Adv. Mater.* **2021**, *33*, 2006323, *Angew. Chem. Int. Ed.* **2020**, *59*, 14935, *Adv. Mater.* **2021**, *33*, 2102134), it

cannot construct a metal-free battery successfully. This experimental result shows the vast obstacles PEO-based electrolytes face in constructing metal-free solid-state batteries, regardless of whether the cathode uses cation or anion hosting materials.

Fig. R3. Columbic efficiency of $\text{LiFePO}_4\text{-Cu}$ batteries with single (up) /dual (bottom) salts.

Thus, we used $50\ \mu\text{m}$ thick Li foil to test the cycle stability of thin lithium battery. During the 250 cycles, the battery did not exhibit any sudden drop in capacity by the failure of the anode.

Fig. S32. PVF cathode cycles with thin Li foil ($50\ \mu\text{m}$).

7) I find quite strange that LiTFSI behaves better than LiFSI. Is there any influence of Fluorine on the performances? Moreover the authors should cite some previous work about Li-salt contribution to SPE performance

a) *Materials Science and Engineering: R: Reports* **134**, 2018, 1-21;

b) *Angew.Chem. Int. Ed.* 2019, **58**,15978-16000

Response:

We thank the reviewer for this suggestion. As the reviewer stated, the importance of fluorine in liquid batteries is widely investigated (*Adv. Mater.* **2018**, *30*, 1706375, *J. Am. Chem. Soc.* **2017**, *139*, 11550). The fluorine-containing electrolyte components, including salts (e.g. LiTFSI, LiFSI) and solvent additives (e.g. FEC), are beneficial to promote LiF formation on the anode side. However, the main factor of the degradation of SPEs-based batteries (especially rate performance) is the ionic conductivity. In this work, since the polymer matrix is PEO, the salts (anions species) directly affect the complexing of ions. LiTFSI is able to dissociate easier than LiFSI, giving higher Li⁺ conductivities in the SPEs. This discussion has been added to the revised manuscript, and the suggested literature has also been cited.

Since the migration of anions and cations in the present work is related to the electrodes' reaction^{43,44}, we set several types of lithium salt in SPEs to obtain a deep insight into the anion electrode reaction.

Ref:

43. Mauger, A., Julien, C. M., Paolella, A., Armand, M. & Zaghbi, K. A comprehensive review of lithium salts and beyond for rechargeable batteries: Progress and perspectives. *Mater. Sci. Eng. R Rep.* **134**, 1-21 (2018).
44. Von Aspern, N., Rösenthaller, G. V., Winter, M. & Cekic-Laskovic, I. Fluorine and lithium: ideal partners for high-performance rechargeable battery electrolytes. *Angew. Chem. Int. Ed.* **58**, 15978-16000 (2019).

8) Does a mix NMC-PVF improve NMC stability with PEO electrolyte? PVF could be used as buffer for other cathodes (NMC, LFP, etc etc)

Response:

We are very grateful to the reviewer for this suggestion. The approach that we introduce in the current paper would enable to use PVF as a modifier for several electrodes as suggested by the reviewer. The classic polymer matrix (PEO) used in this work does not match the high voltage cathode (NMC) well. However, it is reasonable to speculate that the subsequent experimental results can be extended to

similar systems (high-voltage resistant polymers match high-voltage cathode materials). We tested the cyclic stability of the mixed electrode of PVF and LFP using PEO-LiTFSI electrolyte. Interestingly, the addition of PVF has significantly improved the cycle stability of the electrode, and the battery showed no micro-short circuit occurred in 800 cycles at $200 \mu\text{A cm}^{-2}$. However, the specific capacity of the hybrid cathode decreases with the increase of the PVF ratio, which is caused by the lower specific capacity of the PVF. The influence of the PVF ratio can be further observed from the charge-discharge curve. When the PVF ratio is higher (PVF: LFP=1: 1), the curve shows more capacity between 3.35 and 3.75 V, and when the LFP ratio is dominant (PVF: LFP = 1: 2), the curve shows a more obvious LFP discharge platform.

Fig. S29. Cycles of composite cathodes. (a) Capacity and (c) charge/discharge curves of cathode|PEO-LiTFSI|Li (active material (PVF+LFP): conductive agent: binder = 6: 3: 1).

9) What is the possible side reactions related to capacity fading? I believe that these reactions will be much more evident at higher loadings.

Response:

We thank the reviewers for their suggestions. By analyzing the results, the battery capacity decreased slowly and showed a linear decline. No sudden drop that the anode failure might have caused was observed for 4000 cycles (**Figure 1e**), indicating the capacity decrease is mainly due to cathodic side processes. Unlike traditional inorganic cathodes (LFP, LCO, etc.), PVF is a linear polymer synthesized by free-radical polymerization, and the ferrocene group provides its reversible

electrochemical activity. However, its molecular weight is not high ($M_n \approx 4815 \text{ g mol}^{-1}$), and its distribution is wide ($M_w/M_n = 1.708$). Therefore, some of the fragments with small molecular weight might diffuse to the solid electrolytes at the test temperature ($60 \text{ }^\circ\text{C}$), reducing the active electrode materials.

In the case of high mass loading, the organic molecules also tend to dissolve into the same electrolytes, resulting in a poor cycle life (*Materials* **2019**, *12*, 1770). However, this issue could be solved in the follow-up research to determine the optimal composition. Furthermore, designing and constructing high-efficiency host materials makes it possible to achieve physical/chemical adsorption of molecules, thereby avoiding a significant decrease in capacity. Therefore, we believe these strategies could be valuable in using polymers as active materials in solid-state batteries.

10) About Li-metal deposition: the authors should add cross section SEM image and EDS mapping of Li-SPE interface (PVF vs LFP batteries) and cite the following previous works adding a discussion

a) <https://doi.org/10.1016/j.matt.2020.03.015>;

b) *Nano letters* 18 (12), 2018, 7583-7589;

c) *Nano letters* 20 (3), 2019, 1607-1611;

d) *Communications Chemistry* 2, 2019, 131

Response:

We are very grateful to the reviewers for this suggestion. We fully agree with the importance of cross-section analysis for solid-state batteries. To investigate the mechanism of lithium deposition/exfoliation more clearly, we choose high mass loading. However, the premature short-circuit in LFP|Li batteries lead to insignificant changes in the deposition behaviour of lithium metal, only local dendrites. Owing to the uncertainty of the growth position and the opaque nature of the SPEs, the dendrites cannot be easily visualized. Nevertheless, the electrochemical results (voltage drop in **Figure 5c, e**) confirm the conduction of cathode and anode.

The charged battery (entirely deposited lithium) with the mixed cathode (PVF:

LFP=1: 1) after 50 cycles show noticeable results. Based on the initial lithium metal, the newly deposited lithium is smooth and reliable, even if the original lithium is not completely flat. The results prove that the lithium metal behavior leads to the battery's stable cycles. We added the following discussion of this part in the revision and supplemented the relevant literature to support our point.

For the high mass loading tests, pure LFP electrode quickly developed a micro-short circuit during the initial few cycles (Fig. S30) and failed to cycle at a capacity around 1 mAh cm^{-2} . Zaghbi et al. proved by in-situ SEM that dendrites have higher hardness⁵⁸, which is the main reason for the failure of polymer solid-state batteries with conventional cathodes such as NCM and LFP^{30,59,60}. On the other hand, the mixed electrode can maintain a stable cycle only with 50% PVF of active material. The cross SEM of the cycled battery shows that Li^+ can achieve stable and smooth deposition on the lithium metal surface, even if the initial lithium is not ideal (Fig. S31).

Fig. S31. Cross section SEM and EDS mapping results of battery. To obtain the results more clearly, the high-mass loading cathode were prepared to observe the interface of Li-SPEs. The charged battery (entirely deposited lithium) with mixed cathode (PVF: LFP = 1: 1) after 50 cycles show noticeable results.

Ref:

- 30. Cao, D., Sun, X., Li, Q., Natan, A., Xiang, P. & Zhu, H. Lithium dendrite in all-solid-state batteries: growth mechanisms, suppression strategies, and characterizations. *Matter* 3, 57-94 (2020).
- 58. Golozar, M., Hovington, P., Paoletta, A., Bessette, S., Lagacé, M., Bouchard, P. & Zaghbi, K. In situ scanning electron microscopy detection of carbide nature of dendrites in Li-polymer batteries. *Nano Lett.* 18, 7583-7589 (2018).
- 59. Kaboli, S., Demers, H., Paoletta, A., Darwiche, A., Dontigny, M., Clément, D. & Zaghbi, K. Behavior of solid electrolyte in Li-Polymer battery with NMC

cathode via in-situ scanning electron microscopy. *Nano Lett.* **20**, 1607-1613 (2020).

60. Golozar, M., Paoletta, A., Demers, H., Bessette, S., Lagacé, M., Bouchard, P. & Zaghbi, K. In situ observation of solid electrolyte interphase evolution in a lithium metal battery. *Commun. Chem.* **2**, 1-9 (2019).

11) What are other possible polymers with comparable properties? The authors must cite some previous work

a) *Chemical Engineering Journal* 416 (2021) 129171;

b) *Materials* 12 (11), 2019, 1770

Did authors consider the possibility to use other metallocene (cobaltocene)?

Response:

We appreciate the reviewer's valuable suggestion. Previous work has summarized the research progress and application of organic electrodes in detail. We found the reviewer's suggestion on the literature quite useful, and we have added relevant citations in the manuscript.

Sustainability, high capacity, and adjustable redox properties give organic electrodes great potential^{39,40}.

Ref:

39. Zhang, S., Li, Z., Cai, L., Li, Y & Pol, V. G. Enabling safer, ultralong lifespan all-solid-state Li-organic batteries. *Chem. Eng. J.* **416**, 129171. (2021).

40. Mauger, A., Julien, C., Paoletta, A., Armand, M. & Zaghbi, K. Recent progress on organic electrodes materials for rechargeable batteries and supercapacitors. *Materials* **12**, 1770 (2019).

The *p*-type and bipolar polymer, which have anion storage capacity during the redox process, can be used as anion-hosting cathodes. The capacity and voltage can be enhanced through molecular engineering such as group modification or cross-linking, which showed promising results. For example, Héctor et al. used cross-linking phenazine-based polymer to reach 223 mAh g⁻¹ with 3.45 V v.s Li⁺/Li (*ChemSusChem* **2020**, *13*, 2428), comparable to the common inorganic cathodes (LFP, LCO).

We used polyvinyl ferrocene as the cathode in this work. Ferrocene has excellent redox stability. It has been widely used in various energy storage systems in the literature (*Angew. Chem. Int. Ed.* **2014**, *53*, 11036, *ACS Energy Lett.* **2017**, *2*, 869, *J.*

Am. Chem. Soc. **2017**, *139*, 1207). Compared with ferrocene, other metallocenes (Co or Ni as metals atom) are uncommon and much expensive (metal elements). The higher molecular weight reduced the capacity. Therefore, the ferrocene-based polymer is more advantageous than other metallocenes in preparing anion hosting cathode.

Reviewer #3 (Remarks to the Author):

Authors investigate a solid-state dual-ion battery using the typical PEO as polymer matrix and a redox-active polymer as cathode. This system is introduced to avoid the concentration polarization observed with SPEs and provide an active use of the anions, which is significant to the field.

The whole paper is focused on the fact that the improved electrochemical performance of PVF|PEO:LiTFSI|Li is due to the avoidance of concentration gradients. However, there is not enough proof to confirm the hypothesis and therefore further experiments and explanations are needed. This is justified in the following specific as well as general comments:

Response:

We thank the reviewer for the valuable comments. To support the hypothesis/proposed mechanism, we have added several related pieces of literature and provided additional experimental data to the revised version. In addition, we have re-organized our discussion and strengthened our claims, trying our best to address the comments and suggestions fully. Below please find our point-by-point responses.

1. I suggest the authors to mention that this is a dual-ion battery as it is an already in use term to describe this type of battery where anions take place in the reaction. This might help the reader understand the topic.

Response:

Thanks to the reviewers for this suggestion. The description of the battery system is critical, and we very much agree with the reviewer's definition of the dual-ion battery to the system presented in our paper. The original intention of this work is to solve some of the challenges of SPEs based solid-state batteries due to the lack of

effective carriers. To distinguish it from the reported dual-ion gel electrolyte, which focuses on liquid dual-ion batteries' problems, we use the "expand effective carrier" description. We believe this term will be less confusing to the reader. Corresponding changes have been made in the revised version.

The participation of anions in electrode reaction has promoted the development of dual-ion batteries in the liquid or extended gel phase³⁶⁻³⁸.

...

the electrolyte of dual-ion batteries is similar to that of lithium-ion batteries.

...

In this work, to solve the problem of SPEs due to the non-reactive anion migration in SPEs, we build dual-ion solid-state batteries through anion-hosting cathode.

2. Page 3 line 55, what do authors mean by ionizing less free Li⁺? Is it that more ion clusters and aggregates are prone to form? I suggest to clarify

Response:

Thanks to the reviewers for their suggestions, which allows us to improve the manuscript. Lithium-ion is the counter ion of polyanion based single-ion solid polymer electrolytes (SISPEs). However, the strong electrostatic interaction causes the lithium ions to combine with negative charges and render it inactive in the electrochemical reaction, or in other work, the quantity of free Li⁺ is small. This is distinguished from the low number of lithium ions caused by the weak dissociation ability of ion clusters and ions in the polymer-salt system. We made relevant clarifications in the revision.

Though some studies have proved that SISPEs have less strict requirements for ion conductivity^{32,35}, there are still concerns about the number of effective carriers due to strong electrostatic interaction between Li⁺ and the negative charges.

3. Page 3 line 55, authors mention that the cost of using SISPEs is the low ionic conductivity due to the lack of solvation ability. The ionic conductivity of SISPEs is considered low compared to conventional SPE due to the fact that anions do not

contribute to the ionic conductivity, so it is not a fair comparison. In addition, they ascribed it to the lack of solvation ability. The solvation ability is given by the polymer matrix either if it is in a conventional SPE or SISPEs. Further clarification would be beneficial.

Response:

Thank you for the precious suggestion. We agree with the reviewers' views on comparing ion conductivity between SPEs and SISPEs. In the first submission, we did not emphasize that the relatively high ionic conductivity in traditional SPEs cannot fully participate in the electrochemical reaction. In addition, the description of the ionic solvation of the polymer matrix in SISPEs is not objective, especially when it is copolymerized with some flexible segments. We have made further clarifications in the revised version.

By constructing polymer segments with weak interaction with Li^{+33} or grafting anions to the polymer backbone³⁴, SISPEs can achieve a high t_{Li^+} (>0.9). Though some studies have proved that SISPEs have less strict requirements for ion conductivity^{32,35}, there are still concerns about the number of effective carriers due to strong electrostatic interaction between Li^+ and the negative charges.

Ref:

35. Zhu, J., Zhang, Z., Zhao, S., Westover, A. S., Belharouak, I. & Cao, P. F. Single- ion conducting polymer electrolytes for solid- state lithium-metal batteries: design, performance, and challenges. *Adv. Energy Mater.* **11**, 2003836 (2021).

4. Page 5, authors should also provide information about how the T_g changes with the different salts.

Response:

We totally agree with the reviewer suggestion. In **Fig. S5b**, we have added a partially enlarged view of the DSC curve of SPEs, showing the effect of different anion species on T_g . In general, the T_g of SPEs usually reflects the difference in ion mobility. However, in the current study, the link between the ions' mobility and T_g is

not very clear to the system's complexity and other factors that affect the mobility, particularly the anion structure.

Fig. S5. The DSC curves of different electrolytes. After doping LiTFSI and LiBOB, the melting point and melting enthalpy of SPEs decrease significantly.

5. Page 5 line 97. Authors should provide a reference of the method used to calculate the transference number.

Response:

Thanks to the reviewers for their suggestions for supplementing the literature. We have made corresponding supplements in the revised edition.

we measured the lithium-ion transference number (t_{Li^+}) of SPEs through the steady-state current method^{45,46} (results are shown in Fig. S5, Table S2).

Ref:

- 45. Evans, J., Vincent, C. A. & Bruce, P. G. Electrochemical measurement of transference numbers in polymer electrolytes. *Polymer* **28**, 2324-2328 (1987).
- 46. Bruce, P. G. & Vincent, C. A. Steady state current flow in solid binary electrolyte cells. *J. Electroanal. Chem.* **225**, 1-17 (1987).

6. Page 5 line 106. Authors attribute the high charge transfer resistance with PEO-LiBOB to the negatively charged clusters. Why? What about contribution from the different interphases? EIS does not prove that the high Rct is from clusters. Authors should provide other more accurate proof of that.

Response:

Good point. As negative t_{Li^+} transference number often occur in high salt concentration systems, to verify the impact of ion clusters on battery impedance, we changed the salt concentration in PEO-LiBOB (O: Li = 30: 1 and 40: 1) to lead the lithium-ion transfer number to positive (0.13 and 0.16). Even if the ionic conductivity of PEO-LiBOB with low salt concentration is lower than that of the 20: 1 system, the ion motion is dominated by ions (30: 1 and 40: 1) rather than ion clusters (20: 1) show the reduced EIS. This result can well explain the increase in battery impedance in a system dominated by ion clusters. If the resistance dominates the rise in the impedance at the interphases, the resistance of the PEO-LiBOB should increase with increasing the O:Li ratio.

Since negative t_{Li^+} often occur in high salt concentration systems^{48,49}, the low salt concentration was conducted to prove this hypothesis. As shown in Fig. S7, LiBOB-PEO (O: Li = 30: 1, 40: 1) with low ionic conductivity but positive t_{Li^+} show lower resistance.

Fig. S7. Salt concentration impact in PEO-LiBOB. Chronoamperometry profiles and AC impedance spectra before and after polarization (inset) for symmetric Li|Li cells with O: Li at (a) 30: 1 and (b) 40: 1. (c) AC impedance of SPEs, (d) ionic conductivity and t_{Li^+} , (e) EIS plot of PVF|Li with different O: Li.

7. Pate 5 line 107. “Low t_{Li^+} operate poorly” is not correct. PEO, despite its low t_{Li^+} ,

has overall good performance (at least at high temperatures) confirmed by multiple articles and commercial batteries built with this system. I suggest to rephrase it.

Response:

We agree with the reviewer that the way it was written in the first submission is confusing. The original submission implies that the performance is mainly determined by t_{Li^+} only, which is not true, as pointed out by the reviewer. While at low current density, t_{Li^+} may be used to predict the cell performance, this is not the case at high current. The importance of t_{Li^+} will be highlighted when the conductivity is insufficient. We modified the discussion in the revised version to make the statement clearer.

Generally, for a typical Li^+ -hosting cathode, SPE with low t_{Li^+} has strict requirements on ion conductivity, especially under high current density, the scarcity of cations on the electrode surface accelerates the growth of dendrites²².

Ref:

22. Peng, B. A. I., Ju, L. I., Brushett, F. & Bazant, M. Transition of lithium growth mechanisms in liquid electrolytes. *Energy Environ. Sci.* **9**, 3221-3229 (2016).

8. Page 5 line 110. While the ionic conductivity of the SPE will affect the overpotential of the batteries, there are other factors that affect as well. And if the overpotential is controlled by the ion conductivity why do the authors study the anion impact?

Response:

We agree with the reviewer that, in addition to the ionic conductivity, some other factors might affect the cell overpotential, as discussed in the literature with many batteries systems. The motivation to investigate the anion types is not only because of their influence on the cell overpotential but also their effect on the interaction process.

Regarding the influence on the cell overpotential, our experimental results confirmed that the difference in ionic conductivity caused by salts type affected the overpotential of batteries (Fig. S10), calculated from the difference between the redox

peaks ($E_{Ox,p}-E_{Re,p}$) in CV curves.

Fig. S10. The negative correlation between the ionic conductivity and ΔE at 60 °C

We apologize for the insufficiently detailed explanation on electrode potential in the manuscript, which might have confused the reviewers and would confuse many readers. In the manuscript, we use the mean of anodic and cathodic peaks ($(E_{Ox,p}+E_{Re,p})/2$) to represent the electrode potential (E^0). In theory, the difference in SPE conductivity affects both oxidation and reduction processes, and the average of redox peaks should stay the same or have only minimal changes, which is also discussed in **Response 14** about CV results of SPE with/without SN. However, besides the difference in overpotential caused by ionic conductivity, extra changes of E^0 are also observed due to the different anions participating in electrode reaction reactions. This prompts us to explore the impact of the types of anions further.

Another reason to study the anion impact is the related experimental results reported in the literature for liquid electrolyte batteries (*Langmuir* **2004**, *20*, 6631, *J. Phys. Chem. C.* **2010**, *115*, 1985, *J. Phys. Chem. C.* **2011**, *115*, 6775). Yousoo Kim et al. studied the redox properties of ferrocene-terminated alkanethiol self-assembled monolayers (Fc SAM) on the Au surface in the liquid phase (*Nat. Commun* **2020**, *11*, 4194). They demonstrated the dependency of the ferrocene/ferrocenium redox process on the anion type, including ion-pairing with counter anions (Fc-anion) caused by differences in Fc-anion interactions and steric constraints. We wanted to confirm if SPEs follow a similar trend. Therefore, we investigated the impact of the

anion type.

9. Page 6 line 138. Authors use DFT to model the interaction between ferrocenium and anions but then provide statements of what happens in the polymer system: “*the folded long-chain polyvinyl ferrocenium suppress the combination of large anions, weakening the effect of ion-pairing on the electrode potential.*” However, there is no proof of such statement. This section would benefit from further explanations and mentioning more clearly to which experimental results they are referring to. While the difference in experimental electrode potentials is large, the difference from the calculations is very small. In addition, what is the theoretical value? Is the anion the only contributor to the shift in voltage plateau? But before this difference in redox peaks and plateaus was assigned to the difference in ionic conductivity.

Response:

We are very grateful to the reviewers for raising this question, which allows us to provide additional relevant experimental results to discuss the issue further. To prove the influence of steric hindrance on the electrode potential (E_0) generated by the binding energy, we set the linear ferrocene structure (constructed through click-chemistry) as cathode, named **VFS**, which can avoid small molecules (ferrocene) diffusing and deactivating in the electrochemical process. At the same time, it has a significantly lower steric hindrance, lead the effect of binding energy more obvious.

The CV tests using VFS cathode showed that the electrode potential (E_0) with four different type anions is linked to binding energy, matching the computation results except for BOB^- . The E_0 decrease of BOB^- -based reaction is not obvious, which has a specific deviation from binding energy. This result can be attributed to the steric effect of anion-dominated ion clusters. We investigated the BOB^- -based reaction at O: Li ratios of 30: 1 and 40: 1, the E_0 of the VFS cathode is significantly lower than that of the PVF cathode. The above results prove that the steric hindrance of large-volume anions (TFSI^- , BOB^- , FSI^-) with polymer cathode and the ion clusters (BOB^- at 20:1) dominated ion form will both significantly impact the reaction

potential.

The linear ferrocene structure (VFS in Fig. S11a) with lower steric hindrance was tested to clarify this point further. During the CV test with VFS as the cathode (shown in Fig. S11b), the E_0 with all tested anions decreases with the order of calculated BE (Fig. S12a), matching the computation results except for BOB⁻. This can be attributed to the steric effect of anion-dominated ion clusters. The E_0 of the VFS cathode is significantly lower than that of the PVF with O:Li ratios of 30:1 and 40:1 in BOB⁻-based reaction (Fig. S11c, S12b). The above results prove that the steric hindrance of large-volume anions (TFSI⁻, BOB⁻, FSI⁻) and the ion clusters (BOB⁻ at 20:1) both weakens the BE's effect on E_0 .

Fig. S11. Synthesis and electrochemical tests of liner structure ferrocene cathode. (a) synthesis step of VFS, reaction in toluene. **(b, c)** show the CV results with VFS as cathode.

Fig. S12. E_0 comparison with PVF and VFS as cathode. The impact of (a) anion species and (b) the ratio of O: Li in PEO-LiBOB on E_0 .

In this work, the redox unit is ferrocene. Even if used as an internal reference in the liquid system, the electrode potential of ferrocene is affected by the counter ion and solvent. For example, $Fc = 0.40$ V vs SCE (MeCN/[NBu₄][PF₆]) (*Electrochim. Acta* **1973**, *18*, 537.); 0.38 V vs SCE (MeCN/[NEt₄][ClO₄]) (*J. Am. Chem. Soc.* **1974**, *96*, 1087.); 0.51 V vs SCE (glyme/[NBu₄][PF₆]) (*Angew. Chem., Int. Ed.* **1980**, *19*, 640.). In the revised manuscript, the difference in electrode potential E_0 is more obvious, in which the potential of ClO₄⁻ is significantly lower than the remaining three anions. Consequently, the difference in the calculation results is also obvious, respectively -46.05 , -58.92 , -69.84 , -75.37 kcal mol⁻¹. For the electrode potential E_0 , the species of anions and the O: Li determines its value (shown in the above discussion), and the ion conductivity more affects the difference of the redox peak (ΔV) of CV results.

10. Page 6 line 143. The stability of the electrode was examined with LiPF₆, as the anion has a large impact in the electrochemistry, LiTFSI would be a better choice for comparison.

Response:

We thank the referees for their helpful suggestions. We used LiTFSI-TEGDME as the liquid electrolyte to determine the cyclic performance of PVF. The capacity in TEGDME decreases more rapidly, which may be attributed to the better solubility of

PVF in ethers. We have added more discussion about that to the revised version.

we evaluated the capacity and stability of the PVF electrode with liquid electrolyte (1.0 M LiTFSI in TEGDME). As shown in Fig. S14, the capacity increases in the first few cycles, which can be attributed to the cathode electrochemical activation, previously observed in several organic electrodes⁵⁴. The electrode exhibits high initial capacity (107 mAh g⁻¹) and retention, confirming the stable redox property of PVF. Also, the shifts voltage plateau shifts to ~3.27 V when LiPF₆ in EC/DEC was used as the electrolyte compared to ~3.54 for TFSI electrolyte (Fig. S15). Similar differences were also presented by Kim et al. in anions impact of electrode reaction⁵⁵.

Ref:

55. Wong, R. A., Yokota, Y., Wakisaka, M., Inukai, J. & Kim, Y. Probing consequences of anion-dictated electrochemistry on the electrode/monolayer/electrolyte interfacial properties. *Nat. Commun.* **11**, 1-9 (2020).

Fig. S14. Cycle of PVF in 1 M LiTFSI in TEGDME (0.5 C, 1 C = 124 mA g⁻¹).

11. Page 7 line 157. The poor rate performance of LiBOB is assigned to the formation of ions aggregations due to the negative t_+ . However, similar poor performance is observed for LiClO₄ and LiFSI despite them having similar t_+ as LiTFSI and LiFSI having similar ionic conductivity as LiTFSI. Why? Maybe using a different polymer matrix with higher t_+ or different anion mobility would provide better insight on the effect on anion and its mobility.

Response:

We are very grateful to the reviewers for their attention to this issue. The electrolyte ionic conductivity influences the capacity performance in polymer-based solid-state batteries. The three SPEs based on PEO with LiTFSI, LiFSI, and LiClO₄ salts have ionic conductivity of 3.53×10^{-4} , 2.16×10^{-4} , 7.30×10^{-5} S cm⁻¹ at 60°C, respectively. This makes the ratio between the electrolyte conductivities 4.84: 2.96: 1. We believe this large difference in conductivity is responsible for the batteries' performance to a great extent. Besides, as **Response 9** discussed, the ion formation in SPEs should not be ignored.

Before further tests of SPEs, we need to modify the batteries tests on the PEO-LiFSI electrolyte. In the initial manuscript, the charging curve of the PEO-LiFSI batteries jittered under high current density. However, the LSV results did not show a significant oxidation curve. In the first submission, we did not discuss this contradiction in detail and attribute it to possible side reactions of anions with the electrode.

Fig. S18. Charge-discharge curves of solid batteries with FSI anion

We repeated the charge/discharge curve during the revision, and the results showed PEO-LiFSI became stable with strictly controlled H₂O content of SPEs. We believe that the LiFSI tests in the first version of the paper may have interfered with trace amounts of H₂O. In this regard, we strictly re-tested the AC impedance and cyclic voltammetry curves and added error bars to ensure the reliability of the data.

To prove the influence of ionic conductivity on capacity, we changed the temperature to ensure that different anions could work at similar conductivity. By

comparing the batteries performance of PEO-LiTFSI (50 °C) and PEO-LiClO₄ (60 °C) (ionic conductivity ratio: 1.12: 1), PEO-LiFSI (70 °C) and PEO-LiTFSI (60 °C) (ionic conductivity ratio: 1.42: 1), the batteries show similar capacity for all the tested electrolytes. These results indicate that the difference in the ionic conductivity (even less than an order of magnitude) significantly impacts the battery capacity.

Fig. R4. Capacity of anions species with similar ionic conductivity. (a) TFSI⁻ and ClO₄⁻, (b) TFSI⁻ and FSI⁻

12. Page 8. Authors compare PVF with LFP electrodes. However, they have two completely different redox chemistry and surface chemistry. Maybe another redox active polymer that is not *p*-type would be a better choice of comparison. And maybe further comparison with other *p*-type polymers would also benefit this study.

Response:

We agree with the referee's point that comparing different electrode surface reactions is insufficient. As a control sample, we tested n-type organic electrode poly(anthraquinonyl sulfide) (PAQs). The PAQs|Li battery has an obvious micro-short circuit after about 100 cycles under the same testing conditions, and it continues to occur during the subsequent cycles, resulting in a decrease in the coulombic efficiency. This is similar to the result of the LFP|Li battery.

On the other hand, to further prove the advantages of using PVF in the cathode to attract the anion and enhance the performance, we tested the performance of the PVF/LFP mixed electrode. Under the same active material ratio (60 wt%), compared with pure LFP electrodes, the cycle performance of PVF: LFP=1: 1 and 1: 2 mixed

electrodes are much better. These extra results and related discussion have been added to the revised manuscript.

First, we tested an *n*-type organic electrode (Li^+ -hosting) poly(anthraquinonyl sulfide) (PAQs) as a cathode, and the results are similar to those of LFP, with short-circuited appears after a few cycles (Fig. S28). Besides, we also tested a mixture electrode with PVF and LFP, which showed clear cycling improvement, maintaining more than 800 cycles at $200 \mu\text{A cm}^{-2}$ without short circuits (Fig. S29). The above results further proved the effectiveness of expanded carriers to improve the cycle stability.

Fig. S28. Performance of PAQs|Li battery. (a) Synthesis step of poly(anthraquinonyl sulfide) (PAQs) (b) Cycles of PAQs|Li with 1 M LiTFSI in TEGDME. (c) Capacity and (d) charge/discharge curves of PAQs|PEO-LiTFSI|Li.

Fig. S29. Cycles of composite cathodes. (a) Capacity and (c) charge/discharge curves of cathode|PEO-LiTFSI|Li (active material (PVF+LFP): conductive agent: binder = 6: 3: 1).

13. Page 8. Authors study the effect of ionic conductivity. By adding plasticizers, the ionic conductivity improves and so does the battery performance. This would suggest that the reason behind the improved performance is the ionic conductivity but do not give insight on the effect of the anion or the reaction of the PVF.

Response:

As the reviewer pointed out, the results of our work showed the crucial role of ionic conductivity on the battery's performance in any system (including systems with lithium-ion or anion hosting cathodes), especially in SPEs. This work shows that the SPEs conductivity increased by four folds when the plasticizers, SN, was added (8.13×10^{-6} and 2.82×10^{-6} S cm⁻¹ with and without SN, respectively). The results showed that higher ionic conductivity has better battery performance. However, it is worth noting that these ionic conductivities are far lower than most SPEs' reported in the previous literature (usually $\sim 10^{-4}$ S cm⁻¹). The fact that the battery performance in the current work outperforms many other solid-state batteries with more conductive SPEs emphasises the role of expanding the charge carrier that can reduce the requirement of high ion conductivity.

14. Page 8 line 215. Authors provide the CV of the battery with SN additive. It would be interesting to show the CV with the same conditions (temperature and scan rate) without SN in order to compare both systems. With SN the difference between the oxidation and reduction peaks is large, how does that compare to the system without

SN? If it solely depends on the anion as previously discussed it should be the same with or without SN. This could provide additional proof if the improvement is coming from the higher ionic conductivity or the anion.

Response:

We are very grateful for the suggestions. As stated in the **Response 9**, the ionic conductivity significantly influences the ΔV of the CV curve, and the type of anion affects E_0 ($(E_{Ox,p} + E_{Re,p})/2$). We tested the CV curve of the SN-free system under the same conditions and compared it with the SN-content sample. High ionic conductivity corresponds to a low ΔV value. Nevertheless, the difference in the estimated ΔV values in both cases is minimal (3.401 to 3.407 V), indicating no influence of SN and the difference between the oxidation and reduction peaks is controlled mainly by the anions. The result is consistent with the viewpoint in the main part.

Fig. S34. CV results of PVF|Li with 5wt% / without SN in SPEs at 30 °C.

In general, further understanding and investigation of the anion migration should be carried out in order to confirm the hypothesis of this paper. I also suggest the authors to check other papers where they analyze the anion mobility in different systems. LiTFSI is the best performing salt for solid polymer electrolytes for most of the reported systems with different polymer matrices and also cathode materials, therefore if authors want to investigate the effect of the anion migration other polymer systems should be included and not only vary the salt.

Response:

We are very grateful to the reviewers for their valuable suggestions. To fully confirm the hypothesis of this article, we performed tests of mixed cathodes and n-type polymer cathodes. The results well verified our theory: introducing anions to electrode reactions can lead to stable cycles. Regarding the controlled factors of the anion participated electrode reaction, by separately controlling the cathode (low steric active material) and the steric hindrance of the anion (ion cluster), we have a deeper understanding of the anion migration in this system. This part is also benefited from the related work about anion chemistry and analysis of our supplementary experiments.

Due to the better dissociation of lithium ions and stable anion structure, LiTFSI salt shows the broadest range of excellent results in solid-state batteries with traditional lithium-ion hosting cathodes. However, as for the case where cations and anions both participate, ionic conductivity is not the only factor affecting the electrode reaction. The excellent performance TFSI showed has been deeply studied in this work. The conclusion that the cathode and anion influence also provide a reference for further work design and exploration.

As the earliest and most widely studied polymer electrolyte matrix, the research on the physical and chemical properties of PEO has been relatively mature. We chose it to verify the viewpoints of our design in the simplest electrolyte. We very much agree with the reviewer's research suggestions on other electrolyte systems, and we believe that the combination of hybrid cathodes and more advanced electrolytes can be demonstrated in future research work.

REVIEWER COMMENTS

Reviewer #1 (Remarks to the Author):

A very good revision and general work. Should be accepted in present form.

Reviewer #2 (Remarks to the Author):

The authors have well revised the paper and made huge efforts to improve the quality of the paper. The paper is interesting and well written. I recommend a minor revision.

I have just one (important) comment:

I believe that figures S30, R3 and S32 need to be moved to the main text: these graphs are very important to develop a commercial SPE-based Li//LFP battery. A 200um thick lithium metal is not practical and probably 50 um is not practical too. Moreover 1mg/cm² (0.17 mAh/cm²) loading of LFP is not very useful. For a future reader, a capacity that fades after 100 cycles with a loading of 1mAh/cm² is more interesting than stable cycling at 0.17 mAh/cm² loading (my point of view).

Reviewer #3 (Remarks to the Author):

The manuscript has improved after the revision; however, the effect of the anion is still not very clear and some explanations are still missing to support the conclusions and claims. Some examples where further explanations are needed are the following:

1. Page 5 line 97: the high ionic conductivity of PEO-LITFSI (mainly at temperatures below T_m) is explained by the lowest melting point of this system. However, the melting point does not explain the higher conductivity at temperatures below the melting temperature, instead the T_g would have a more pronounced effect and it is higher for this system. So a different explanation is needed.

2. Page 5 line 111 and Figure S7: at which temperature are these experiments? This is important information to compare with the other salt concentration. PEO-LiBOB with O:Li 30:1 and 40:1 although they show "lower" ionic conductivity (can't be compared because temperature is not reported) they have lower resistance and higher t₊, which might indicate better SPE materials, why lower salt concentration has not been used?

3. Page 6 line 124: "This indicates that the ion pair (ferrocenium-anion) cannot undergo further reduction." why it cannot undergo further reduction? And if PEO-LiClO₄ has shown the worst performance why is it used for the reference example?

4. Page 6 line 124: "This indicates that the ion pair (ferrocenium-anion) cannot undergo further reduction. In contrast, PVF with its long-chain structure prevents the diffusion of active materials by anchoring ion pairs into the polymer" these two sentences are talking about different things. Is the fact that ferrocenium-ClO₄ does not reduce solved by using

PVF instead? If the polymer is used to avoid the diffusion of active material the reference should have shown a continuous capacity fade. In this reference the ferrocenium-ClO₄ is not reversible but it is with PVF-PEO-LiClO₄. Why? Furthermore, the diffusion of active material could be significant with liquid electrolytes but it should be more limited with SPEs.

5. Page 6 and fig 4: the focus is on the potential difference, but the shape of the CV is also different depending on the anion. Nothing about that is mentioned. Furthermore, in my opinion, the explanation of the E₀, BE and anion could benefit from clearer explanations.

6. Temperatures of every experiments should be clearly marked in the text and figure captions.

Response to Reviewers' Comments

Reviewer #1 (Remarks to the Author):

A very good revision and general work. Should be accepted in present form.

Response:

We thank the reviewer for the positive comment and support.

Reviewer #2 (Remarks to the Author):

The authors have well revised the paper and made huge efforts to improve the quality of the paper. The paper is interesting and well written. I recommend a minor revision.

I have just one (important) comment:

I believe that figures S30, R3 and S32 need to be moved to the main text: these graphs are very important to develop a commercial SPE-based Li//LFP battery. A 200um thick lithium metal is not practical and probably 50 um is not practical too. Moreover 1mg/cm² (0.17 mAh/cm²) loading of LFP is not very useful. For a future reader, a capacity that fades after 100 cycles with a loading of 1mAh/cm² is more interesting than stable cycling at 0.17 mAh/cm² loading (my point of view).

Response:

We thank the reviewer for the positive comment and support. We fully agree with the referee's concern about figure arrangement. In order to avoid irrelevant data in the article, we removed the results (50 μm thickness Li foil) that cannot effectively reflect the stability of lithium metal. In response to the mass load issue raised by the referee, we have emphasized the performance of high loading electrodes and made additional discussion to the revision.

Page 8, line 213

Importantly, for the high mass loading tests, pure LFP electrode quickly developed a micro-short circuit during the initial few cycles (Fig. 5e, S30) and failed to work at a capacity around 1 mAh cm⁻². In contrast, mixed cathode (mass ratio, LFP: PVF = 1: 1) showed clear cycling improvement, maintaining more than 90 cycles without short circuits.

Fig. 5. Carrier expansion improves rate performance and cycle stability. Schematic diagram of anode morphology changes during cycling where (a) PVF and (b) LiFePO₄ served as the cathode. (c) The voltage-time curves of PVF|Li (top) and LFP|Li (bottom) batteries at 300 μA cm⁻². (d) Coulombic efficiency of PVF|Li battery over 4000 cycles. (e) Cycles with LFP and mixed cathode (mass ratio = 1:1) at high loading. (f) The rate performance from 100 to 1000 μA cm⁻² and (g) the comparison between this work and the other reported advanced SPEs, classified in the graph by design strategy (detail seen in Table S5). All the tests were performed at 60 °C

Reviewer #3 (Remarks to the Author):

The manuscript has improved after the revision; however, the effect of the anion is still not very clear and some explanations are still missing to support the conclusions and claims.

Some examples where further explanations are needed are the following:

1. Page 5 line 97: the high ionic conductivity of PEO-LiTFSI (mainly at temperatures below T_m) is explained by the lowest melting point of this system. However, the melting point does not explain the higher conductivity at temperatures below the melting temperature, instead the T_g would have a more pronounced effect and it is higher for this system. So a different explanation is needed.

Response:

We thank the reviewer for pointing out this problem to help us improve the article's reliability. As the referee said, the ionic conductivity does not correspond well with the DSC results. The polymer-ion aggregation forms led by anions are different, and the chain segment movement can not be directly linked to the ion motion. Ionic conductivity should result from the combined effects of the physical properties of the polymer (glass transition temperature, melting point, etc.) and the ionic structure within it. We have explained this in the revised version.

Page 5, line 95

The DSC results show that, relative to pure PEO, both SPEs exhibit decreased T_m and produce glass transition processes, implying enhanced segmental motion. LiTFSI and LiBOB, which exhibit high ionic conductivity, have a lower degree of crystallinity (determined by the melting peak intensity). The T_m and T_g changes with anions species are not directly related to the ionic conductivity. Different anions impact the polymer-salt composite structure, resulting in various degrees of segment motion. Meanwhile, salt dissociation affects the conductivity. The negative charge delocalization of anion helps release more free ions.

2. Page 5 line 111 and Figure S7: at which temperature are these experiments? This is important information to compare with the other salt concentration. PEO-LiBOB with O: Li 30:1 and 40:1 although they show “lower” ionic conductivity (can’t be compared because temperature is not reported) they have lower resistance and higher t_{Li^+} , which might indicate better SPE materials, why lower salt concentration has not been used?

Response:

We thank the reviewers for their suggestions. During the entire tests of the PEO-LiBOB system, the temperature was set at 60 °C. We have supplemented descriptions in the corresponding texts and modified the figures. At tested temperature (60 °C), SPEs with low LiBOB concentration can report better ion migration numbers, but the ionic conductivity shows a decreasing trend. Meanwhile, the lower LiBOB concentration, with low ionic conductivity and positive t_{Li^+} , is expected to show similar behavior to other salts (e.g., LiClO₄). The effect of ion clusters induced by high LiBOB concentration (O: Li at 20: 1) on the electrochemical performance is also interesting and worthy of further study. Thus, we conducted the O: Li at 20:1 in PEO-LiBOB to perform the tests.

Fig. S7. Salt concentration impact in PEO-LiBOB. Chronoamperometry profiles and AC impedance spectra for symmetric Li|Li cells with O: Li at (a) 30:1 and (b) 40: 1. (c) AC impedance, (d) ionic conductivity and t_{Li^+} of SPEs. (e)

EIS plot of PVF|Li with different O: Li in PEO-LiBOB. All the electrochemical tests were performed at 60°C.

3. Page 6 line 124: “*This indicates that the ion pair (ferrocenium-anion) cannot undergo further reduction.*” why it cannot undergo further reduction? And if PEO-LiClO₄ has shown the worst performance why is it used for the reference example?

Response:

Thanks to the reviewer for this suggestion. For the redox of small-molecule organic electrode (e.g., ferrocene) in batteries, the active materials diffusion problem is very prominent, especially in liquid systems. In the replacement system with SPEs, the test temperature is usually close to or higher than the melting point of the electrolyte, leading to a strong fluidity of the polymer segment. In this case, the small-molecule cathode material tends to undergo diffusion and deactivation, similar to the liquid system. On the other hand, compared with the liquid system, the electron/ion transport network between the electrode molecules and the electrolyte is poorer, and the mutual independence of small molecules makes the electrode reversible failure aggravated. Therefore, the ferrocenium-anion ion pair cannot be further reduced after the initial oxidation. We have supplemented the relevant explanations in the original text.

Fig. R1. Schematic diagram of ferrocene redox failure in SPEs.

Although LiClO₄ performs poorly in capacity and rate, mainly due to its lower ionic conductivity. However, the CV results of PVF|Li with LiClO₄ as salt exhibit good reversibility, same as other salts (shown in **Fig. 4b**). PEO-LiClO₄ has lower

segment mobility (owing to its high crystallinity) among the tested system. We selected it as a comparison to realize the reversible redox of small molecule electrodes in SPEs as much as possible. However, it turns out that even adopting SPE with low segment motion, diffusional deactivation of small-molecule electrode materials still exists.

4. Page 6 line 124: “This indicates that the ion pair (ferrocenium-anion) cannot undergo further reduction. In contrast, PVF with its long-chain structure prevents the diffusion of active materials by anchoring ion pairs into the polymer” these two sentences are talking about different things. Is the fact that ferrocenium-ClO₄ does not reduce solved by using PVF instead? If the polymer is used to avoid the diffusion of active material the reference should have shown a continuous capacity fade. In this reference the ferrocenium-ClO₄ is not reversible but it is with PVF-PEO-LiClO₄. Why? Furthermore, the diffusion of active material could be significant with liquid electrolytes but it should be more limited with SPEs.

Response:

We apologize for making it confuse here. The CV results with PEO-LiClO₄ (**Fig. R1**) show that the ferrocene cannot undergo a reversible reduction process after the initial oxidation peak. In contrast, the PVF cathode exhibits reversible redox peaks in the continuous CV tests, proving the effectiveness of replacing ferrocene with PVF.

Fig. R2. Comparison of CV results with (a) small molecules, ferrocene and (b) polymer, PVF as cathodes. The electrolyte is PEO-LiClO₄, and the test temperature is 60°C.

As stated in *Response 3*, the realization of reversible redox processes in PVF is mainly attributed to avoiding the small molecules' diffusion and charge transport problem. To further clarify this point, we used PEO-LiTFSI matched ferrocene to test the redox properties at 60 °C. As shown in **Fig. R3**, the first oxidation peak shows a lower intensity due to the stronger segment mobility of PEO-LiTFSI. Meanwhile, during the charge-discharge process, the battery exhibits low capacity only at the initial charge, and the subsequent capacity is negligible. The strong diffusivity of small molecules leads to irreversible redox. Furthermore, the small-molecule electrodes severely reduced the electron/ion transport between the polymer electrolyte and the active material, which seriously hindered the electrochemical process of ferrocene in SPEs. We have supplemented a detailed explanation of the issue in our revised version.

Fig. R3. (a) CV results and (b) charge-discharge curves of Ferrocene|PEO-LiTFSI|Li batteries, tested at 60 °C

Page 6, line 127

The solid-state battery with the ferrocene cathode (Fig. S9) exhibits an oxidation peak only in the first cycle. This can be attributed to the ion pairs diffusion into the SPEs, owing to its segment motion at high temperature. In addition, the poor electron/ion transfer interface between the small molecule cathode and the electrolyte further hinders the redox process. In contrast, PVF with anchored active units can maintain well reversible redox.

The dissolution-diffusion of organic electrodes in liquid electrolytes is evident.

Although SPEs are less mobile than liquids, however, above the melting point, the motion of the polymer chains still puts the electrode material at risk of diffusion. Specifically, currently reported organic molecule-based solid-state batteries mainly adopt inorganic solid electrolytes (ISEs) to avoid cathode loss (*Angew. Chem. Int. Ed.*, **2018**, *57*, 8567, *ACS Energy Lett.*, **2021**, *6*, 201-207, *Joule*, **2019**, *3*, 1349-1359). Thus, as stated by referees, the diffusion of active material could be significant with liquid electrolytes but cannot be easily ignored in SPEs.

5. Page 6 and fig 4: the focus is on the potential difference, but the shape of the CV is also different depending on the anion. Nothing about that is mentioned. Furthermore, in my opinion, the explanation of the E_0 , BE and anion could benefit from clearer explanations.

Response:

We thank the reviewers for their suggestions, which were significant in improving the article. As stated by the reviewer, the CV shape correlates with the anion species. In the TFSI-involved CV, the primary redox process was followed by a recessive weak peak. However, the redox peaks with different shapes also have good reproducibility, which indicates that the process is also reversible. In liquid systems, CVs of ferrocene exhibit non-ideal behavior, commonly observed in close-packed Fc self-assembled monolayers (SAMs) (*Langmuir*, **2006**, *22*, 4438-4444, *J. Phys. Chem. C.*, **2015**, *119*, 21978-21991, *J. Phys. Chem. C.*, **2013**, *117*, 1006-1012). These can be attributed to reasons including local heterogeneity and intermolecular interactions.

The close packing of ferrocene units is also involved in this work, and the splitting and shape difference of CV peaks should be related to this. Taking TFSI as an example, multiple redox waves (or so-called peak splitting) appear in the CV peak when PVF is used as the cathode. However, in the CV results of VFS cathodes where the ferrocene units are relatively dispersed, there is no apparent peak separation, which fully proves that the aggregation morphology of the active cathode units has a significant effect on the CV shape. For the anions that are bulky or have special strongly electronegative atoms (F atoms), the buried active unit (ferrocene) can lead

to multiple electron transfer planes (PETs), leading to non-ideal CV behavior (*J. Phys. Chem. C*, **2015**, *119*, 21978-21991, *Langmuir*, **2006**, *22*, 4438-4444, *J. Phys. Chem. B*, **2001**, *105*, 9557-9568).

Fig. R3. Schematic diagram and results of CV curve shape change. **(a)** The binding process of anions to active units while PVF (left) and VFS (right) are used as cathodes, respectively. **(b)** The stacking of active units differentiates the binding process of anions, leading to the splitting of CV peaks. **(c)** No obvious fractals are observed in the CV results with the VFS as cathode.

On the other hand, ion differences are also expected to affect cathode-cathode and cathode-anion interactions, further affecting CV shape (*J. Phys. Chem. C*, **2011**, *115*, 1985-1995, *Nat. Comm.*, **2020**, *11*, 4194, *Langmuir*, **2018**, *34*, 1327-1339). Supplementary explanations were provided in the revised Supporting Information.

Supplementary Note 1: Shape of cyclic voltammograms with PVF cathode

All CVs in main text **Fig. 4b** correspond to reversible redox. The repeatability suggests that impurities are unlikely to be the source of the non-ideal behavior. The CV of an ideal surface binding reaction contains a single peak. However, some anion (typically, TFSI⁻) pairs exhibit deviations from ideal behavior, with asymmetry/multimodality and peak broadening. There are

theoretically potential origins for the anion-related properties of peak shape, including:

(1) Buried Fc and plane of electron transfer (PET). Active unit (ferrocene) packing in PVF buries part of the Fc. As described in the previous experiments and model about close-packed Fc self-assembled monolayers (SAMs) (*Anal. Chem.*, **1992**, *64*, 2398-2405, *J. Phys. Chem. C*, **2015**, *119*, 21978-21991), the positional difference between the active unit and the closest ion/ion cluster creates multiple electron transfer planes (PETs) that induce splitting and broadening of the CV peak. We noticed that making the units farther apart (VFS cathode with TFSI⁻ anion, shown in **Fig. S11**) obtained a more ideal CV.

(2) Intermolecular interactions experienced by the Fc and anions. For Fc surface-bound redox activity, CVs can be fitted to models based on Langmuir or Frumkin isotherms to gain insight into the nature of intermolecular interactions (*J. Am. Chem. Soc.* **2016**, *138*, 9611-9619, *Nat. Comm.*, **2017**, *8*, 2066), including oxidized and reduced forms of Fc with anions. In addition, differences in anion size make internal transport potentially sterically hindered, leading to discriminatory CV results (*Langmuir*, **2018**, *34*, 1327-1339).

In addition, we thank for the referee's suggestion about the explanation, and revised manuscript to make the information clearer.

Page 6, line 137

The results show that the ion pairs' formation could negative shift the electrode potential (E_0) from theoretical, affected by binding capability^{52,53}. We calculate the binding energy (BE) of ion pairs by density functional theory (DFT) simulations. The cathode was simplified by substituting ethyl ferrocenium for the PVF (Table S4). As Fig. 4a shows, the ethyl ferrocenium has the highest BE to ClO₄⁻, and decrease with the order FSI⁻, BOB⁻, and TFSI⁻. However, in the CV results of PVF, the E_0 with TFSI⁻, BOB⁻, and FSI⁻ as anions have no significantly differences (3.463, 3.470, and 3.476 V, respectively). The discrepancy between the experimental and computational results can be explained by the steric hindrance of

active units (ferrocene) and ions/ion clusters: (1) The folded long chain in PVF inhibits the binding of large anion to ferrocene, thus, reducing the BE effect on E_0 . When using VFS (Fig. S11a), with low steric hindrance for ferrocene, as cathode, the E_0 of TFSI, FSI decreased significantly (Fig. S12a), matching the trend of the calculated results; (2) The anion-dominated ion clusters exhibit a larger steric structure, which enhances the hindrance effect. In PEO-LiBOB, the change in salt concentration resulted in different aggregation morphologies of ions, as previously described (Fig. 3b, S7). Compared with O: Li at 20: 1, the decrease of E_0 (PVF to VFS) is more evident in low salt concentration (30: 1, 40: 1), shown in Fig. S11c, S12b. The results demonstrate that, apart from the cathode, the steric hindrance on anion side also weakens the E_0 drop caused by the binding process. Therefore, avoiding ion clusters, ClO_4^- , with the smallest size (Fig. S13) and the strongest BE (Fig. 4c), shifts E_0 to more negative values (3.381 V vs. Li^+/Li), while the other anions were not significantly affected by the binding effect (Fig. 4d).

6. Temperatures of every experiments should be clearly marked in the text and figure captions.

Response:

Thanks to the reviewer for the suggestion. We have carefully reviewed all experimental sections of the manuscript, including supporting information, and added the description of the test temperatures.

REVIEWERS' COMMENTS

Reviewer #2 (Remarks to the Author):

I recommend to accept the paper in the present form

Reviewer #3 (Remarks to the Author):

The authors have done a good job revising and improving the manuscript. It could be accepted.

Response to Reviewers' Comments

Reviewer #2 (Remarks to the Author):

I recommend to accept the paper in the present form.

Response:

We are grateful for the reviewer's recommendation.

Reviewer #3 (Remarks to the Author):

The authors have done a good job revising and improving the manuscript. It could be accepted.

Response:

Thank you very much for your great encouragement to our work.